# miRNAs-Set of Plasmatic Extracellular Vesicles as Novel Biomarkers for Hepatocellular Carcinoma Diagnosis Across Tumor Stage and Etiologies

**DOI:** 10.3390/ijms26062563

**Published:** 2025-03-12

**Authors:** Francisco A. Molina-Pelayo, David Zarate-Lopez, Rosendo García-Carrillo, César Rodríguez-Beas, Ramón Íñiguez-Palomares, José L. Rodríguez-Mejía, Adriana Soto-Guzmán, Gabriela Velasco-Loyden, Mónica Sierra-Martínez, Adolfo Virgen-Ortiz, Enrique Sánchez-Pastor, Nancy E. Magaña-Vergara, Rafael Baltiérrez-Hoyos, Javier Alamilla, Victoria Chagoya de Sánchez, Adán Dagnino-Acosta, Enrique Chávez, Luis Castro-Sánchez

**Affiliations:** 1Centro Universitario de Investigaciones Biomédicas, Universidad de Colima, Colima 28045, Colima, Mexico; franciscoalfredo_molina@ucol.mx (F.A.M.-P.); dzarate@ucol.mx (D.Z.-L.); rosendo_garcia@ucol.mx (R.G.-C.); pv_jrodriguez@ucol.mx (J.L.R.-M.); avirgen@ucol.mx (A.V.-O.); espastor@ucol.mx (E.S.-P.); javier_alamilla@ucol.mx (J.A.); dagninoa@ucol.mx (A.D.-A.); 2Departamento de Física, Universidad de Sonora, Hermosillo 83000, Sonora, Mexico; cesar.rodriguez@unison.mx (C.R.-B.); ramon.iniguez@unison.mx (R.Í.-P.); 3Departamento de Medicina y Ciencias de la Salud, Universidad de Sonora, Hermosillo 83000, Sonora, Mexico; adriana.soto@unison.mx; 4Departamento de Biología Celular y Desarrollo, Instituto de Fisiología Celular, Universidad Nacional Autónoma de México, Ciudad de México 04510, Mexico; nvelasco@ifc.unam.mx (G.V.-L.); vchagoya@ifc.unam.mx (V.C.d.S.); 5Unidad de investigación en Salud, Hospital Regional de Alta Especialidad de Ixtapaluca, Servicios de Salud del Instituto Mexicano del Seguro Social para el Bienestar (IMSS-BIENESTAR), Ciudad de México 01020, Mexico; monica.sierram@hraei.gob.mx; 6Facultad de Ciencias Químicas, Universidad de Colima, Coquimatlán 28400, Colima, Mexico; nancymv@ucol.mx; 7SECIHTI—Universidad de Colima, Colima 28045, Colima, Mexico; 8SECIHTI—Universidad Autónoma Benito Juárez de Oaxaca, Oaxaca 68120, Oaxaca, Mexico; rbaltierrez@hotmail.com

**Keywords:** microRNAs, extracellular vesicles, biomarkers, diagnosis, hepatocellular carcinoma

## Abstract

Hepatocellular carcinoma (HCC) is the most common primary liver cancer, often diagnosed at advanced stages due to insufficient early screening and monitoring. MicroRNAs (miRNAs) are key regulators of gene expression and potential biomarkers for cancer diagnosis. This study investigated the diagnostic potential of miRNAs in Extracellular Vesicles (EVs) from HCC. miRNA expression in EVs was analyzed using HCC cell lines, circulating EVs from a Diethylnitrosamine (DEN)-induced liver tumor rat model, and plasma samples from HCC patients. Receiver Operating Characteristics (ROCs) were applied to evaluate the diagnostic accuracy of circulating EV miRNAs in patients. Five miRNAs (miR-183-5p, miR-19a-3p, miR-148b-3p, miR-34a-5p, and miR-215-5p) were consistently up-regulated in EVs across in vitro and in vivo HCC models. These miRNAs showed statistically significant differences in HCC patients stratified by TNM staging and Edmondson–Steiner grading compared to healthy controls. They also differentiated HCC patients with various etiologies from the control group and distinguished HCC patients, with or without liver cirrhosis, from cirrhotic and healthy individuals. Individually and as a panel, they demonstrated high sensitivity, specificity, and accuracy in identifying HCC patients. Their consistent upregulation across models and clinical samples highlights their robustness as biomarkers for HCC diagnosis, offering the potential for early disease management and prognosis.

^‡^ These authors contributed equally to this work and shared senior authorship.

## 1. Introduction

Liver cancer comprises several hepatic neoplasms [1], and hepatocellular carcinoma (HCC) is the most common type, accounting for approximately 80% of cases [2], and is considered the third leading cause of cancer-related mortality [3].

The lack of early diagnosis is a significant factor contributing to high mortality, considering the low detection rate and the rapid progression of the tumor stages. Imaging techniques combined with serum markers, such as ultrasound testing with serum alpha-fetoprotein determination [4], constitute a standard diagnostic panel for HCC; however, deficient early screening and monitoring, as well as limited timely detection programs, significantly delay diagnosis [5]. The detection of abnormal liver mass by computed tomography and magnetic resonance imaging is commonly used for monitoring cirrhotic patients, and these methods may be sufficient to confirm HCC [6]. However, in cases where imaging is inconclusive, a liver biopsy and histological analysis have been recommended for confirmation [7], and sometimes, up to three biopsies are necessary to establish a conclusive HCC diagnosis [8].

The HCC prognosis is known to be unfavorable due to rapid tumor growth, although early detection increases life expectancy [9]. However, the HCC diagnosis is mainly made at the advanced stage, which reduces treatment options and estimated survival of patients [10]. Therefore, HCC patients urgently need novel diagnostic methods for screening and monitoring that are non-invasive, sensitive, specific, and relatively easy to use.

Accordingly, previous studies have proposed serum/plasma biomarkers as a new diagnostic strategy, including circulating levels of tumor proteins and epigenetic markers such as noncoding RNAs [11,12]. In addition to soluble factors, cells in the tumor microenvironment secrete vesicle-enclosed signals in a nonclassical mechanism to regulate intercellular communication. Extracellular Vesicles (EVs) are cell-derived secretory vesicles originating from multivesicular endosomal bodies or the plasma membrane [13]. EV cargo in cancer cells has been reported to include oncogenes, signaling and extracellular matrix proteins, RNA transcripts, and microRNAs (miRNAs), suggesting that they may influence the tumor microenvironment [14].

miRNAs are a class of short noncoding RNAs that alter posttranscriptional and translational mRNA targets to regulate various functions such as cell proliferation, differentiation, migration, and apoptosis, as well as pathological processes, including cancer [15,16,17]. Previously, impaired expression of miRNAs has been associated with the tumorigenesis, invasion, and metastasis of HCC [18,19,20], and several miRNAs have been proposed as biomarkers for HCC. Moreover, EVs released by cancer cells change their content depending on cancer development, cell origin, and tumor microenvironment [21], and they could be a potentially sensitive diagnostic tool. This study aimed to analyze the expression of miRNAs associated with EVs in different HCC cell lines and a DEN-induced liver tumor rat model to characterize their differential expression according to cancer progression stages and validate their sensitivity and specificity as potential biomarkers in HCC patients. We identified a five-panel of circulating miRNAs with > 99% sensitivity and specificity for a novel diagnostic test for HCC patients.

## 2. Results

### 2.1. Isolation and Characterization of EVs Secreted by HCC Cell Lines

Non-tumoral liver cell line (THLE-2) and five hepatoma cell lines subclassified by differentiation stages, two well-differentiated (PLC/PRF/5 and C3A-HepG2) and three poorly differentiated (SNU-449, SNU-475, and SNU-423), were cultured. EVs from cultured cell lines were isolated through sequential ultracentrifugation and characterized by their physical and molecular properties. The EV morphology analyzed by TEM exhibited a quasi-spherical shape for EVs derived from non-tumoral and tumoral cells (Figure 1A and B, respectively). Also, the DLS analysis showed the presence of two size-distributed EV subpopulations, with a range from 30 ηm to 700 ηm and a peak at the size of 40–100 ηm (Polydispersity Index (PDI), 0.428 to 0.709) in the first subpopulation and 200–700 ηm (PDI, 0.415 to 0.724) in the second subpopulation (Figure 1C,D, Appendix A). According to the International Society of Extracellular Vesicles (ISEV) guidelines, these two subpopulations can be referred to as small (first peak) and medium/large EVs (second peak). Furthermore, EVs produced by hepatic cell lines showed the presence of EV markers Alix, Hsp90 α/β, Flotilin-1, and Tsg101 and the absence of the negative marker calnexin (Figure 1E). Additionally, the average protein content in EV-enriched fractions from tumoral cells (5.23 ± 1.16 mg/mL) exhibited an increase greater than four-fold compared to non-tumoral THLE-2-EVs (Figure 1F). In addition, the RNA concentration in EVs secreted specifically by PLC/PRF/5 exhibited two-fold more RNA (9.80 ± 0.624 ng/mL) than non-tumoral EVs. Interestingly, the RNA content in EVs from the other tumoral cell lines evaluated was similar (6.94 ± 1.79 ng/mL) to non-tumoral THLE-2-derived EVs (Figure 1G). Therefore, these findings suggest that changes in the RNA expression profile, rather than RNA concentrations, may play a role in EV-derived signaling from cancer cell lines.

### 2.2. miRNA Expression Profiles of EVs Derived from Hepatoma Cell Lines

To analyze the cell and EV miRNA profiles, we performed a microarray to detect 84 cancer-related miRNAs. The heat maps showed several up-regulated and down-regulated miRNAs in hepatoma cell lines and their produced EVs compared to non-tumoral THLE-2 cells and EVs, respectively (Figure 1H,I). Although miRNA expression patterns in HCC cell lines were not related to their differentiation stage (Figure 1H), the miRNAs contained in EVs from poorly differentiated HCC cell lines showed an upregulation compared to THLE-2, suggesting a relationship with the differentiation stage (Figure 1I). Also, we examined the miRNA expression profiles exclusively found in HCC cell lines and their EVs using Venn diagrams (Figure 2A and B, respectively). Ten miRNAs were uniquely expressed in PLC/PRF/5, four in C3A (HepG2), one in SNU-449, and ten in SNU-475. Moreover, miR-183-5p was the only common miRNA expressed in all HCC cell lines (Figure 2A and Appendix A). Only SNU-449 and SNU-423 from EVs displayed seven and one uniquely expressed miRNAs, respectively (Figure 2B and Appendix A). miR-183-5p, miR-19a-3p, miR-148b-3p, miR-34a-5p, and miR-215-5p were common miRNAs expressed in EVs secreted from hepatoma cell lines (Figure 2B). To confirm these results, we evaluated the miRNA expression by RT-qPCR (Figure 2C,D). Among the five miRNAs commonly expressed in EVs, only miR-183-5p was detected in all HCC cells. In contrast, miR-19a-3p, miR-148b-3p, miR-34a-5p, and miR-215-5p were differentially expressed in some tumoral cell lines (Figure 2C). Interestingly, these five miRNAs showed higher expression in EVs derived from HCC cell lines (Figure 2D), suggesting that they are significantly modulated when they are contained in EVs.

### 2.3. miRNA Expression Profiles in Circulating Plasma EVs of DEN-Induced Liver Tumor Rat Model

To determine the expression levels of miR-183-5p, miR-19a-3p, miR-148b-3p, miR-34a-5p, and miR-215-5p in circulating plasma EVs, we developed a DEN-induced liver tumor rat model. First, we evaluated liver injury markers in plasma (ALT and GGT) and hepatic tissue (lipid peroxidation), HCC markers in liver biopsies (Afp and Gpc3 expression), and morphometric analysis in both groups (Figure 3A,B and Appendix A). Liver injury markers revealed a twofold increase in plasmatic ALT concentrations and no changes in GGT levels; however, we found a fivefold increase in hepatic lipid peroxidation in DEN-treated rats compared with control rats (Appendix A). Morphometric analysis showed the presence of nodules and tumors in the liver tissue of DEN-treated rats (Figure 3A). Also, the Afp and Gpc3 expression levels in hepatic tissue increased more than two-fold and four-fold, respectively, and the histopathological analysis of liver biopsies confirmed the development of liver tumors in DEN-treated rats (Figure 3A,B and Appendix A). Once the liver tumor model was validated, we purified plasmatic EVs by sequential ultracentrifugation and examined the EV markers by Western blot and particle size and distribution by TEM and DLS. Protein levels of Alix, Hsp90 α/β, Flotilin-1, and Tsg101 in circulating EVs were similar between control and DEN-treated rats (Figure 3C). TEM analysis exhibited a quasi-spherical shape for EVs derived from the plasma of control and DEN-treated rats (Figure 3D). The DLS analysis revealed three distinct subpopulations of EVs in DEN-treated rats, which had different sizes ranging from 16 ηm to 338 ηm. The size distribution of these three EVs peaked at 16–24 ηm, 36–67 ηm, and 208–338 ηm, while the control rats exhibited a unique population of EVs ranging from 629 to 762 ηm (Figure 3E,F and Appendix A). These subpopulations of EVs can be classified as small (first peak) and medium/large EVs (second and third peak) in DEN-treated rats and only medium/large EVs in control rats. In addition, the protein and RNA levels in plasma-circulating EVs did not significantly differ in both groups (*p* = 0.488 and *p* = 0.068, respectively) (Figure 3G,H). Next, we studied the expression levels of miRNAs by RT-qPCR in liver tissue and plasma-circulating EVs (Appendix A). The relative expression of these five miRNAs did not display differences in hepatic miRNA expression in DEN-treated rats compared to control rats (*p* = 0.49, *p* = 0.07, *p* = 0.59, *p* = 0.59, and *p* = 0.17, respectively) (Appendix A). Interestingly, their expression levels were higher in plasma-circulating EVs obtained from DEN-treated rats compared with control rats (*p* = 0.005, *p* = 0.0014, *p* = 0.003, *p* = 0.036, and *p* = 0.0016, respectively) (Figure 3I). Furthermore, the most significant change in expression was for miR-183-5p, miR-19a-3p, and miR-148b-3p (Figure 3I). These results suggest a potential application of these five miRNAs as HCC biomarkers.

### 2.4. miRNA Expression Assessment in Plasma EVs Obtained from HCC Patients

We recruited 90 HCC patients with different clinicopathological features and 41 healthy volunteers without liver pathologies. First, we confirmed liver injury (AST and ALT) and HCC markers (AFP) in the plasma of both groups and reported the clinical pathological features of HCC patients (Appendix A). Next, we purified plasma-circulating EVs and analyzed the expression levels of the five candidate miRNAs by RT-qPCR to confirm our previous results (Figure 4). Our results demonstrated that RNA concentration in plasma EVs was significantly higher in HCC patients compared to healthy subjects (*p* < 0.0001) (Appendix A). In addition, the five miRNAs evaluated showed a higher expression in circulating plasma EVs from HCC patients compared to control subjects (*p* < 0.0001) (Figure 4A–E). Moreover, we assessed miRNA expression data by classifying patients according to HCC etiologies (HBV, HCV, ethanol, and unknown), as well as the presence or absence of cirrhosis (Figure 5A,B). The five miRNAs showed significant upregulation in HCC patients across all etiologies compared to healthy individuals (*p* < 0.0001), but no significant differences were observed among the different etiologies analyzed (*p* > 0.05) (Figure 5A). Interestingly, the miRNA set did not distinguish healthy controls from cirrhotic patients (Figure 5B). Furthermore, the miRNAs differentiated HCC patients with and without cirrhosis from both healthy individuals and cirrhotic patients. (*p* < 0.0001); however, no significant differences were observed between cirrhotic and non-cirrhotic HCC patients (*p* > 0.05) (Figure 5B). Additionally, we analyzed the miRNA expression levels in plasmatic EVs across different TNM stages (I to IV) and Edmondson–Steiner grades (I-II to III-IV) of HCC (Figure 6). The five candidate miRNAs showed significant upregulation in all TNM stages (Figure 6A–E, right panels) and Edmondson–Steiner grades (Figure 6A–E, left panels) compared to healthy subjects (*p* < 0.0001). Remarkably, the expression levels of all analyzed miRNAs effectively differentiated HCC patients of any tumoral stage and grade from healthy subjects (Figure 6). Moreover, the expression levels of miR-183-5p, miR-148b-3p, miR-34a-5p, and miR-215-5p in EVs from patients stratified by TNM staging and Edmondson–Steiner grading exhibited an overall upward trend as tumor stage or grade progressed (Figure 6). However, in most TNM classification stages, they did not show a statistically significant difference (*p* > 0.05). In contrast, in the Edmondson–Steiner grading system, the expression levels of miR-183-5p, miR-148b-3p, and miR-34a-5p (Figure 6A,C,D, left panels) displayed a significant difference across grades (*p* < 0.0001). Finally, miR-34a-5p was significantly increased in TNM stage III and IV compared to stage I of HCC (*p* = 0.049 and *p* = 0.027, respectively) and miR-148b-3p in TNM stage IV compared to stage I (*p* = 0.008) (Figure 6 C,D, right panels). Overall, the results suggest a possible clinical application of these five candidate miRNAs contained in EVs (miR-183-5p, miR-34a-5p, miR-148b-3p, miR-19a-3p, and miR-215-5p).

### 2.5. Efficacy Validation of miRNA Contained in Plasma-Circulating EVs as Diagnostic Biomarkers of HCC

To validate the diagnostic accuracy of miR-183-5p, miR-34a-5p, miR-148b-3p, miR-19a-3p, and miR-215-5p as HCC biomarkers, we evaluated the ROC curves for each miRNA in HCC patients compared to a control group (Appendix A and Table A1, part I and II). ROC analysis revealed that the AUC for each miRNA was the following: 0.9233 for miR-183-5p (95% CI, 0.879–0.967, *p* < 0.0001); 0.9224 for miR-19a-3p (95% CI, 0.877–0.967, *p* < 0.0001); 0.9576 for miR-183-5p (95% CI, 0.926–0.989, *p* < 0.0001); and 0.9115 for miR-34a-5p (95% CI, 0.861–0.961, *p* < 0.0001) (Table A1, part I and II). Furthermore, the miRNAs analyzed had a sensitivity of 84.44% to 94.44%, a specificity of 82.93% to 95.12%, an accuracy of 83.68% to 88.56%, a Youden’s Index of 0.6737 to 0.7981, a positive likelihood ratio (LR+) in an interval between 5 to 17, and a negative likelihood ratio (LR-) close to zero. Furthermore, miR-148b-3p showed the highest sensitivity (94.44%) and accuracy (89.90%), while miR-215-5p presented the maximum LR+ (17.303) and specificity (95.12%) (Table A1, part I and II). These results suggested that the five candidate miRNAs provided promising metrics for discriminating HCC patients from healthy subjects. In addition, we performed the ROC analysis for every miRNA in HCC patients stratified by TNM staging and Edmondson–Steiner grading compared with the control group (Table A1 and Table A2 and Appendix A). The results showed that the five candidate miRNAs had an interval of sensitivity 80.65% to 95.45%, specificity of 75.61% to 97.56%, accuracy of 78.28% to 94.21%, Youden’s Index of 0.5656 to 0.8743, LR+ of 3.318 to 35.393, and LR- close to zero in HCC patients classified by the TNM system. Moreover, the evaluation of diagnostic accuracy parameters of EV-derived miRNAs in HCC patients stratified by Edmondson–Steiner grades revealed sensitivity ranging from 79.41% to 92.86%, specificity of 73.17% to 95.12%, accuracy of 80.00% to 93.26%, Youden’s Index of 0.6141 to 0.875, and LR+ of 3.29 to 87.50, while LR- was close to zero. Also, these results indicated that among the five miRNAs analyzed, miR-148b-3p and miR-215-5p had the most influential parameters in HCC patients by TNM staging, while miR-183-5p and miR-215-5p exhibited the best parameters in HCC patients by Edmondson–Steiner grading. However, we were unable to discriminate between specific stages or grades of HCC. Despite this limitation, the five candidate miRNAs successfully distinguished HCC patients stratified by TNM staging and Edmondson–Steiner grading from healthy subjects (Table A1 and Table A2).

### 2.6. miRNA Contained in Plasma-Circulating EVs as a Diagnostic Panel for HCC Detection

To increase the diagnostic accuracy of the analyzed miRNAs, we determined the optimal correlation between miRNA expression by combiROC (Appendix A). The analysis exhibited several miRNA arrangements to obtain the best diagnostic panel for categorizing HCC patients of all TNM stages from healthy subjects (Appendix A). These results showed that increasing the number of miRNAs (two to five) included in each combination improves the accuracy in AUC, sensitivity, and specificity metrics (Table A1 and Table A2 and Appendix A). Thus, combining the five candidate miRNAs was the most suitable for a diagnostic panel to discriminate HCC patients in all tumoral stages with AUC = 1 and sensitivity, specificity, and accuracy of 100% (Appendix A). Moreover, for HCC stages III and IV, the optimal number of miRNAs combined in a diagnostic panel is reduced to three with similar results (Appendix A). The findings suggest that the five candidate miRNAs analyzed provided a novel diagnostic panel for detecting HCC across tumor stages, grades, and etiologies.

## 3. Discussion

In this study, we examined miRNA expressions in EVs from tumoral liver cell lines, plasma samples of a DEN-induced liver tumor model in rats, and HCC patient plasma. Our findings revealed consistent miRNA expression patterns in EVs, which may be helpful in HCC diagnosis and prognosis.

The physical and molecular properties of EVs were similar across hepatoma tumor cell lines. We identified small and medium-sized EV populations based on established classification criteria and detected EV-specific markers, including Flotilin-1 and Hsp90 α/β, in all cell lines [22]. Variations in Alix and Tsg-101 expression may reflect heterogeneity in the exosome biogenesis process [23].

We observed differences in EV total protein content but not EV RNA content across all hepatoma cell lines compared to the non-tumoral THLE-2 line. While previous studies have reported proteomic and RNA profile differences in carcinoma cells, these variations have not been linked to total concentrations [24,25,26]. Despite similar total RNA content, distinct RNA expression profiles were evident, as confirmed by differential miRNA expression in EVs, and the selective sorting of miRNAs into EVs is associated with cancer progression and metastasis [27]. Notably, no consistent miRNA expression pattern was observed in total cell lines; however, a clear distinction emerged between well-differentiated and poorly differentiated tumor cells. Previous studies have identified 167 differentially expressed EV miRNAs in hepatoma cell lines, targeting genes involved in key cancer pathways such as proliferation, angiogenesis, and metastasis [28].

In our study, more than 50 deregulated EV miRNAs were found, and to refine the association of miRNA expression with HCC, we searched for commonly deregulated miRNAs across all tumor cell lines and identified miR-183-5p as the only miRNA consistently expressed in total cell samples from all HCC cell lines. In contrast, five miRNAs, including miR-183-5p, were consistently expressed in EVs from all HCC cell lines and showed differential expression compared to the non-tumoral cell line. Previous studies have described miRNA transfer from cells to exosomes as a mechanism of intercellular communication [29,30] in normal cells and as a modulator of the tumor microenvironment in cancer cells [31].

To validate these findings, we developed a DEN-induced liver tumor model in rats to assess miRNA expression in circulating EVs. Plasma-derived EVs were purified, revealing three size-based subpopulations. However, EV sizes were comparable across all tumor cell lines. The presence of multiple EV subpopulations may reflect the greater heterogeneity of EVs in a living system compared to cell lines [32]. Additionally, the diverse EV subpopulations identified in DEN-treated rats may be linked to the pleiotropic effects of tumor cell-derived EVs [33,34,35].

We examined the expression of the five miRNAs commonly found in EVs from HCC hepatoma cell lines and plasma-derived EVs from a DEN-induced liver tumor model. Interestingly, their expression levels remained similar in total hepatic tissue but were elevated in purified EVs compared to healthy rats. Previous studies suggest that cancer progression and metastasis may be influenced by EV-mediated signaling in hepatocellular carcinoma [27,36]. Several miRNAs have been identified as exclusive or differentially expressed in HEpG2 and PLC/PRF/5 hepatoma cell-derived EVs, playing roles in cell growth, viability [25], proliferation [37], and metastasis [38]. In addition, miRNAs carried by circulating EVs can modulate the tumor microenvironment and affect distant tissues [31,39,40,41]. Given these findings, miRNAs in circulating EVs have been proposed as more precise biomarkers for cancer detection and progression [40].

We confirmed the elevated expression of the same five EV-contained miRNAs in HCC patient samples, stratified by TNM stage, Edmondson–Steiner grade, etiologies, and hepatic cirrhosis, compared to healthy subjects. These findings strongly support the potential of miRNAs in plasma-derived EVs as biomarkers for HCC detection. Previous research has linked increased miR-92b expression in circulating EVs to early HCC recurrence after transplantation [41]. Moreover, miR-665, miR-224, and miR-638 levels in EVs have been associated with tumor size and disease stage in HCC patients [42,43,44]. A recent study also identified five miRNAs (hsa-let-7a, hsa-miR-21, hsa-miR-125a, hsa-miR-200a, and hsa-miR-150) in fucosylated serum EVs that distinguished HCC patients, stratified by the BCLC staging system, from non-HCC controls [45]. Supporting our results, miR-215-5p was found to be elevated in circulating EVs from HCC patients [46], while serum exosomal miR-34a was significantly down-regulated, although the specific expression of its 3p or 5p arm was not determined [47]. Collectively, these findings, including our study, underscore the potential of EVs as a valuable source of biomarkers for HCC diagnosis.

The five miRNAs analyzed in this study were significantly up-regulated in HCC patients across all etiologies compared to healthy individuals; however, no statistically significant differences were observed between etiologies. Previous investigations have reported elevated levels of miR-222 and miR-223 in the serum of HCC patients, regardless of HBV status, compared to healthy controls [48]. Additionally, five plasma EV-derived miRNAs (miR-19-3p, miR-16-5p, miR-223-3p, miR-30d-5p, and miR-451a) were significantly increased in HCC patients without hepatitis B or C compared to healthy subjects, though this study did not analyze HBV- or HCV-associated HCC [49]. Furthermore, plasma levels of miR-21-5p and miR-155-5p were notably up-regulated in HCC patients with HCV etiology compared to cirrhotic patients, although healthy individuals were not included in the comparison [50].

We performed ROC analysis to evaluate the diagnostic potential of these five miRNAs in HCC patients, stratified by TNM stage and Edmondson–Steiner grade, compared to healthy individuals. Previous studies have identified EV-derived miRNAs with high diagnostic accuracy for distinguishing HCC from healthy individuals. Notably, miR-10b-5p exhibited the highest AUC of 0.932 in a cohort of 90 HCC patients compared to 28 healthy individuals, highlighting the strong discriminatory power of EV miRNAs in HCC diagnosis [46]. A study involving 72 HCC patients demonstrated that serum EV miRNAs had higher diagnostic performance than the same miRNAs freely circulating in plasma for distinguishing HCC from HBV or cirrhotic patients [51]. Additionally, another study with 80 patients reported an AUC of 0.850 for miR-17-5p in serum-derived EVs, with increased miR-106a expression strongly correlating with poor survival [52]. However, some of these studies did not account for differences between tumor stages or assess diagnostic performance in distinguishing HCC from healthy individuals. In contrast, our study showed that most of the analyzed miRNAs effectively discriminated HCC even when patients were stratified by tumor stage, further supporting their potential as diagnostic biomarkers.

Previous research has demonstrated that combining multiple miRNAs enhances diagnostic accuracy. For instance, a three-miRNA ratio signature exhibited high diagnostic accuracy in differentiating HCC patients from non-HCC controls [45]. Interestingly, a computational screening of differentially expressed liver miRNAs across three large HCC datasets further demonstrated improved diagnostic performance using a two-miRNA combination in a combiROC analysis [53]. Compared to a single miRNA with the highest AUC for distinguishing HCC from healthy patients (miR-215-5p; AUC 0.9642), a combiROC analysis using two miRNAs (miR-215-5p/miR-19a-3p) increased the AUC to 0.992, achieving maximum accuracy with a five-miRNA panel contained in EVs. Previous studies have similarly reported improved diagnostic performance using EV miRNA combinations. For instance, a serum EV miRNA combiROC analysis increased the AUC to 0.780 in a cohort of 24 HCC patients compared to 17 healthy controls, while a plasma EV miRNA combiROC analysis raised the AUC to 0.85 in 48 HCC patients versus 20 healthy individuals [54]. However, it is crucial to recognize that combining ROC parameters for multiple markers does not inherently improve diagnostic performance, as the right combination of biomarkers is necessary to achieve high accuracy. For example, in a study involving 40 HCC patients with liver cirrhosis, the combiROC analysis of miR-199a with miR-21-5p resulted in a lower AUC for differentiating HCC from cirrhosis compared to the combined AUC of AFP and miR-21-5p. Similarly, the miR-199a/miR-155-5p combination had the same AUC as miR-155-5p alone, emphasizing the need for careful biomarker selection to enhance diagnostic precision [50].

Our findings suggest that the five-miRNA panel identified in EVs from hepatoma cell lines, as well as in circulating EVs from the DEN-induced liver tumor model and patients with different HCC stages, grades, and etiologies, may represent a novel strategy for screening and monitoring, facilitating HCC diagnosis.

## 4. Materials and Methods

### 4.1. Cell Culture

The cell lines used were classified as the control human liver cell line THLE-2, and the tumoral HCC cell lines were subclassified as well-differentiated (PLC/PRF/5 and C3A-HepG2) and poorly differentiated (SNU-449, SNU-475, and SNU-423), these cell lines were obtained from the American Type Culture Collection (ATCC, Manassas, VA, USA). THLE-2 cells were grown in DMEM/F12 (Gibco, Thermo Fisher Scientific, MA, USA), while well-differentiated HCC tumor cell lines were grown in MEM (11095080, Gibco, Thermo Fisher Scientific, MA, USA) and poorly differentiated in RPMI-1640 (11875093, Gibco, Thermo Fisher Scientific, MA, USA). For all conditions, cells were supplemented with 10% FBS (26140079, Gibco, Thermo Fisher Scientific, MA, USA) and antibiotics (100 µg/mL penicillin and streptomycin and 10 µg/mL gentamicin). Culture conditions were maintained in 10 cm culture plates in a humidified incubator (37 °C) and 5% CO_2_.

### 4.2. Isolation of Extracellular Vesicles

EVs were obtained from the culture supernatant of liver non-tumoral and tumoral HCC cell lines, plasma samples of a DEN-induced liver tumor rat model, and plasma samples from healthy volunteers and HCC patients. Cells were grown (2 × 10^6^) on a 10 cm plate in 10% FBS-supplemented culture medium (26140079, Gibco, Thermo Fisher Scientific, MA, USA) to 80% confluence and subsequently placed in Exo-Depleted serum (A2720801, Gibco, Thermo Fisher Scientific, MA, USA) and incubated for 48h to avoid FBS exosome contamination. The culture supernatants and 1 mL of plasma samples were collected immediately and centrifuged at 2000× *g* (30 min) and subsequently at 12,000× *g* (30 min) to remove cell detritus. EVs were isolated by ultracentrifugation (acceleration rate from 0 to 500 rpm in 1 min and deceleration rate from 500 to 0 in 6 min) at 100,000× *g* for 3 h. All centrifugation cycles were performed at 4 °C. Purified EVs were processed for the intended specific analysis (Appendix A).

### 4.3. Hepatocellular Carcinoma-Induced Rat Model

Male adult Wistar rats (Crl:WI) weighing 250 to 300 g were acquired (Charls River Laboratories, NC, USA) and housed at the Animal Facility of the Universidad Nacional Autónoma de México. All animal experiments were conducted following the Mexican federal regulations NOM-ZOO-1999-062 [55] regarding the protection of animals used for scientific purposes and received approval by the Ethics Committee for the Care and Handling of Laboratory Animals of the Universidad Nacional Autónoma de México with national authorization SAGARPA-SENASICA AUT-B-C-1216-030 (permission number of institutional protocol approval CICUAL-VCHH156-19). All procedures were approved and supervised, and this study is reported following the ARRIVE guidelines. The animals were fed ad libitum and housed under controlled conditions (22 ± 2 °C, 50–60% relative humidity, and 12 h light–dark cycles). To induce HCC, rats selected by simple randomization and housed in separated rooms were injected intraperitoneally with DEN (50 mg/kg) once a week for 16 weeks (*n* = 10) and maintained without DEN injection for an additional six weeks as recovery time. Rats were sacrificed 22 weeks after the first DEN injection (*n* = 5, death percentage 50%) and liver injury and HCC markers in serum and hepatic tissue were evaluated. Alanine aminotransferase (ALT) levels were determined by an ALT activity assay (MAK052, Merck, Darmstadt, Germany), and gamma-glutamyl transferase (GGT) was evaluated by a fluorometric GGT activity assay kit (MAK089, Merck, Darmstadt, Germany), according to the manufacturer’s specifications. Lipid peroxidation (mmol MDA/mg protein) was determined by measuring malondialdehyde (MDA) formation using the thiobarbituric acid method previously described by Ohkawa et al. [56].

As described in the previous sections, *Afp* and *Gpc3* expression in hepatic rat tissue was determined by RT-qPCR. Morphometric analysis determined body, liver, and spleen weight and the liver/body weight ratio. Finally, liver tissue samples were processed for histological analysis using the hematoxylin and eosin technique.

### 4.4. Ethical Approval, Informed Consent for Participation, and Sample Collection from Human Subjects

This case–control study comprised 90 patients with clinical and histopathological diagnoses of hepatocellular carcinoma at different tumor stages or grades who underwent percutaneous liver biopsy and had not received antiviral or antitumoral treatment or undergone liver surgery, 10 cirrhotic patients, and 41 healthy control subjects, as described in Appendix A. All HCC patients were included based on tumor stage (TNM classification) or grade (Edmondson–Steiner system) and the presence of steatosis, necroinflammation, and cirrhosis, and then biopsy results were classified. The cirrhotic patients were evaluated using clinical and biochemical studies, as well as the imaging test for HCC discarding. The healthy volunteers were subjected to various biochemical studies to verify that they did not present metabolic or liver function alterations; the control subjects were not infected with hepatitis B or C virus and were not alcohol consumers (Appendix A). Patients included were individuals with at least three generations of residence in Mexico, aged between 30 and 70 years, and diagnosed with HCC that had undergone liver tissue biopsy along with clinical and imaging studies to determine tumoral stage or grade and have a record of full medical history. Exclusion criteria considered patients with antiviral or antitumoral treatment, renal or cardiac disease, lipid and/or carbohydrate metabolism disorders, pregnancy, immunological diseases, or mental disabilities preventing informed consent. Also, patients were eliminated from the study when primary tumors were confirmed outside the liver by biopsy or all required research procedures could not be completed. Peripheral venous blood samples were collected in a sodium citrate Vacutainer tube (BD, Franklin Lakes, NJ, USA) and centrifugated at 1500× *g* by 15 min to separate plasma from the globular package. This study was conducted in accordance with the Declaration of Helsinki, and all human samples were collected following informed consent. Participants were allowed to withdraw voluntarily at any stage, and the study was terminated accordingly. All procedures were approved and supervised by the local ethics committees of Juarez Hospital of Mexico (approval no. CEI-HJM-P/014/2015) and Ixtapaluca High Specialty Regional Hospital (approval no. NR-014-2022). This study was reported following the STARD guidelines for reporting diagnostic accuracy studies [57].

### 4.5. Data Analysis

According to the normal distribution, data are expressed as the mean ± standard deviation (SD) or median ± interquartile range (IQR). One-way analysis of variance (ANOVA) with Dunnett’s post hoc test was employed for multiple comparisons. Two-tailed Student’s t-tests were used for two unpaired sample comparisons. For relative miRNA expression in patients, a two-tailed Mann–Whitney U test for two unpaired sample comparisons and Kruskal–Wallis with Dunn’s post hoc test for multiple comparisons were applied. Statistical significance was set at 95% (*p* <0.05). Prism software v.9.5.1 (GraphPad, San Diego, CA, USA) and Origin Pro v.9.95 (OriginLab Corporation, Northampton, MA, USA) were used for statistical analysis.

Receiver operating characteristics (ROCs) were applied to miRNA expression data to differentiate HCC patients from control non-tumoral patients. The ROC parameters considered were the area under the curve (AUC), sensitivity, specificity, and accuracy. In addition, a combiROC analysis was conducted by combining the cut-off values of each candidate miRNA.

## 5. Conclusions

Our findings indicate that five miRNAs (miR-19a-3p, miR-34a-5p, miR-148b-3p, miR-183-5p, and miR-215-5p) contained in EVs are differentially expressed in hepatoma cell lines compared to non-tumor liver cells, suggesting their potential as HCC biomarkers. We confirmed their expression in circulating EVs in both a DEN-induced liver tumor rat model and in plasma samples from HCC patients with various tumor stages, grades, and etiologies. Each of these five EV-derived miRNAs demonstrated high diagnostic performance for discriminating HCC, and their combined analysis further increases their potential as biomarkers. Finally, we recognize as a limitation of our study that the miRNA set only discriminates healthy individuals from HCC patients. Therefore, future research should evaluate this miRNA set in other liver pathologies. Moreover, prospective cohort studies remain essential for confirming the effectiveness of these biomarkers in early detection under clinical conditions.

## Figures and Tables

**Figure 1 ijms-26-02563-f001:**
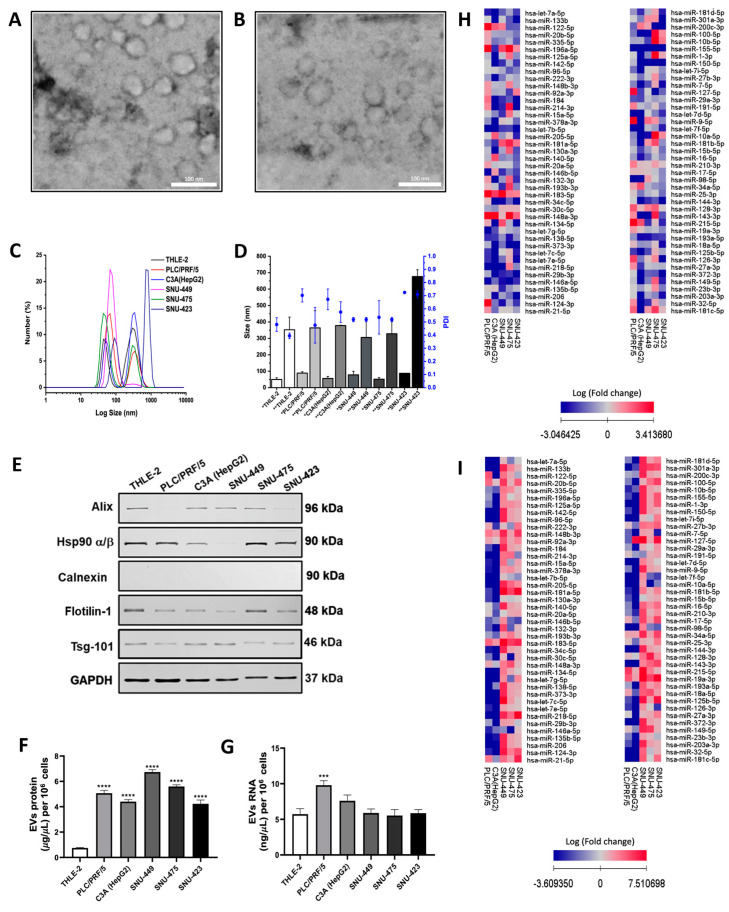
Physical and molecular characterization of HCC cell-derived EVs. The EVs purified by the ultracentrifugation of culture supernatants of non-tumoral and HCC cell lines were evaluated by TEM and DLS. The representative microphotographs show the morphology of EVs secreted by THLE-2 (**A**) and PLC/PRF/5 (**B**) cell lines. DLS analysis for EV subpopulation size-distribution (**C**) and PDI (**D**). The PDI values are represented by blue points over each bar (* peak 1 and ** peak 2) with the scale on the right side of the graph. Total RNAs and protein extracts were obtained from liver cell-derived EVs. The EV markers (50 mg protein/lane) were analyzed by Western blot (**E**). A representative image for Western blot is shown. The protein (**F**) and RNA (**G**) of EVs secreted per 1 × 10^6^ cells were measured by Bradford assay and fluorometry, respectively. Real-time qPCR miRNA microarrays *miScript miRNA Array Human Cancer PathwayFinder kit* (331221 MIHS-102ZD, Qiagen, Hilden, Germany) were effectuated using 300 ng of the RNA of cells and EVs. The heat maps reveal the miRNA expression for cells (**H**) and their secreted EVs (**I**). The miRNA amounts were normalized with U6 levels (ΔCt). The miRNA expressions were calculated as relative expressions (2^-ΔΔCt^). The values in the heat maps for tumoral cells or EVs represent the fold change (logarithmic scale) to non-tumoral THLE-2 cells and EVs. The up-regulated and down-regulated miRNAs are indicated by squares colored with a scale of red to blue, respectively. Data are means ± SD in DLS analysis or protein and RNA quantification of at least three independent experiments. *** *p* < 0.001 and **** *p* < 0.0001 vs. THLE-2 cells or EVs.

**Figure 2 ijms-26-02563-f002:**
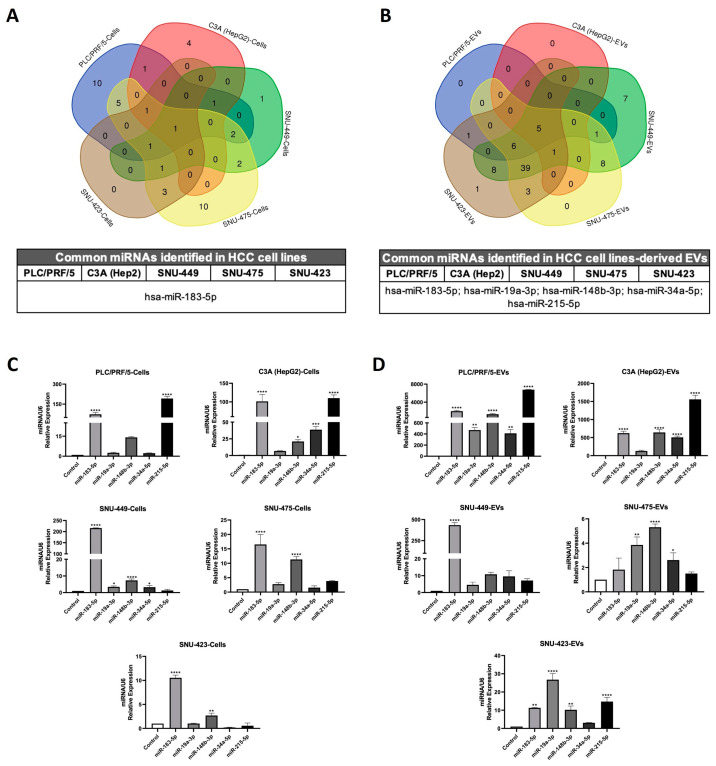
Microarray validation by RT-qPCR of shared miRNAs identified in HCC cell lines and their secreted EVs. The up-regulated miRNAs (≥2-fold change vs. THLE-2) of the HCC cells (**A**) and their secreted EVs (**B**) obtained from microarray data analysis were evaluated by Venn diagrams. The expression of common miRNAs in hepatoma cells (**C**) and EVs (**D**) was determined by RT-qPCR. The miRNA amounts were normalized with U6 levels (ΔCt). The miRNA expression was calculated as relative expression (2^−ΔΔCt^) to non-tumoral THLE-2 cells or their EVs. Data are means ± SD of at least three independent experiments. * *p* < 0.05, ** *p* < 0.01, *** *p* < 0.001, and **** *p* < 0.0001 vs. THLE-2 cells or EVs.

**Figure 3 ijms-26-02563-f003:**
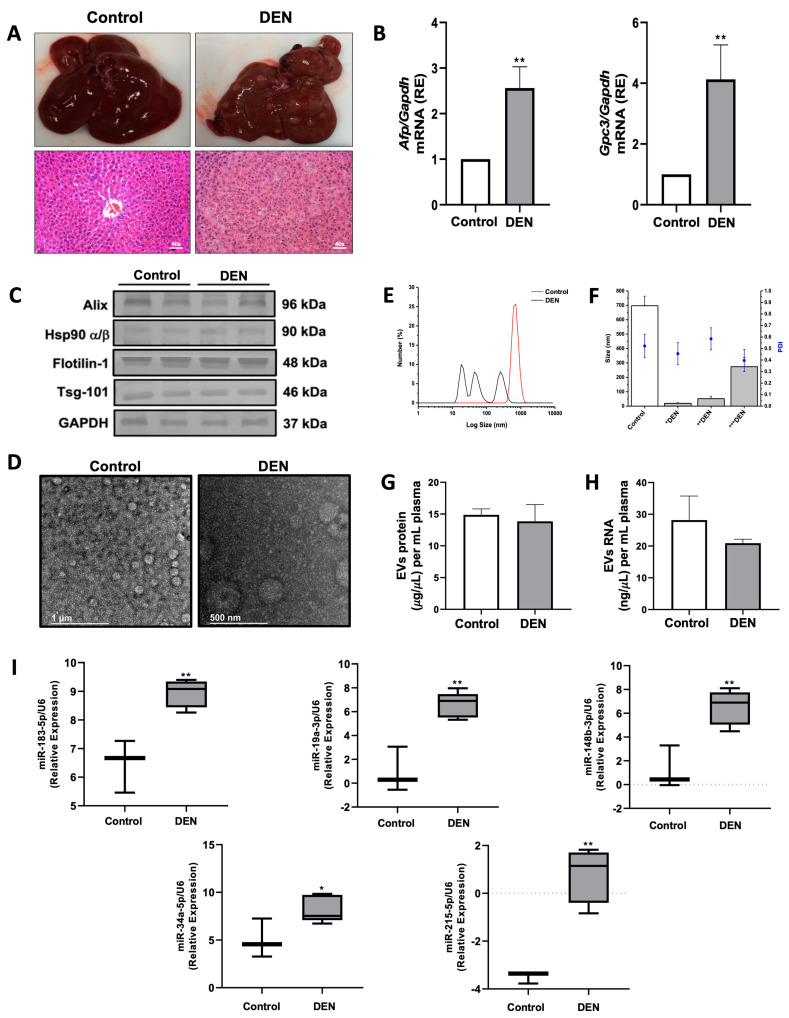
miRNA expression measurement in plasma EVs of DEN-induced liver tumor rat model. To induce liver tumors in rats, they were injected i.p. with DEN (50 mg/kg) once a week for 16 weeks (*n* = 10). The rats were euthanized at 22 weeks after the first DEN injection, and the liver and plasma were obtained for further analysis (*n* = 5, death percentage = 50%). Hepatic tissue of control and DEN-treated rats ((**A**), upper left and right panels) and histological analysis of H&E stained slices from liver biopsies ((**A**), lower left and right panels). Total RNAs were obtained from liver tissue in both analyzed groups. Afp and Gpc3 mRNA expression levels in hepatic tissue by RT-qPCR (**B**). The mRNA amounts were normalized with Gapdh levels (ΔCt). The mRNA expressions of DEN-treated rats were calculated as relative expression (2^−ΔΔCt^) to control rats. EVs were purified by ultracentrifugation of the plasma DEN-induced liver tumor rat model. The EV markers (50 mg protein/lane) were evaluated by Western blot (**C**) and the morphology by TEM (**D**). Representative images for Western blot and microphotography are shown. DLS analysis for plasmatic EV subpopulation size-distribution (**E**) and PDI (**F**). The PDI values are represented by blue points over each bar (* peak 1, ** peak 2, *** peak3) with the scale on the right side of the graph. Total RNA and protein extracts were obtained from plasmatic EVs for control (*n* = 3) and DEN-treated rats (*n* = 5). The protein (**G**) and RNA (**H**) of circulating EVs were measured by Bradford assay and fluorometry. The miRNA expressions in EVs of control and DEN-treated rats were determined by RT-qPCR (**I**). Both groups calculated miRNA expressions as relative expressions and normalized U6 levels. Data are means ± SD for mRNA expression, DLS analysis, RNA, and protein quantification in at least three independent experiments. The box plot graphs represent the median and interquartile ranges for miRNA data expression. * *p* < 0.05 and ** *p* < 0.01 vs. control group.

**Figure 4 ijms-26-02563-f004:**
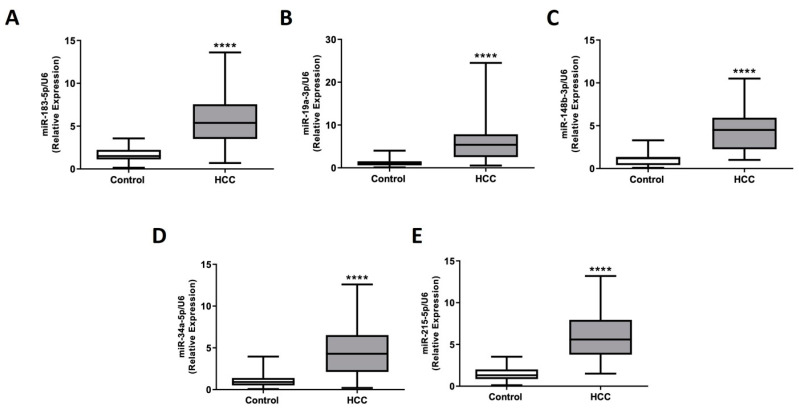
Evaluation of miRNA expression in plasma-circulating EVs of HCC patients. EVs were purified by ultracentrifugation of the plasma of HCC patients with different clinicopathological features (*n* = 90) and healthy subjects without hepatic pathologies (*n* = 41). Total RNAs were obtained from the circulating EVs of plasma patients. The miRNA expressions in control subjects and HCC patients were determined by RT-qPCR (**A**–**E**). The miRNA expressions were calculated as relative expressions and normalized U6 levels in both patient groups. miRNA data expressions are represented in box plot graphs indicating the median and interquartile ranges. **** *p* < 0.0001 vs. healthy subjects.

**Figure 5 ijms-26-02563-f005:**
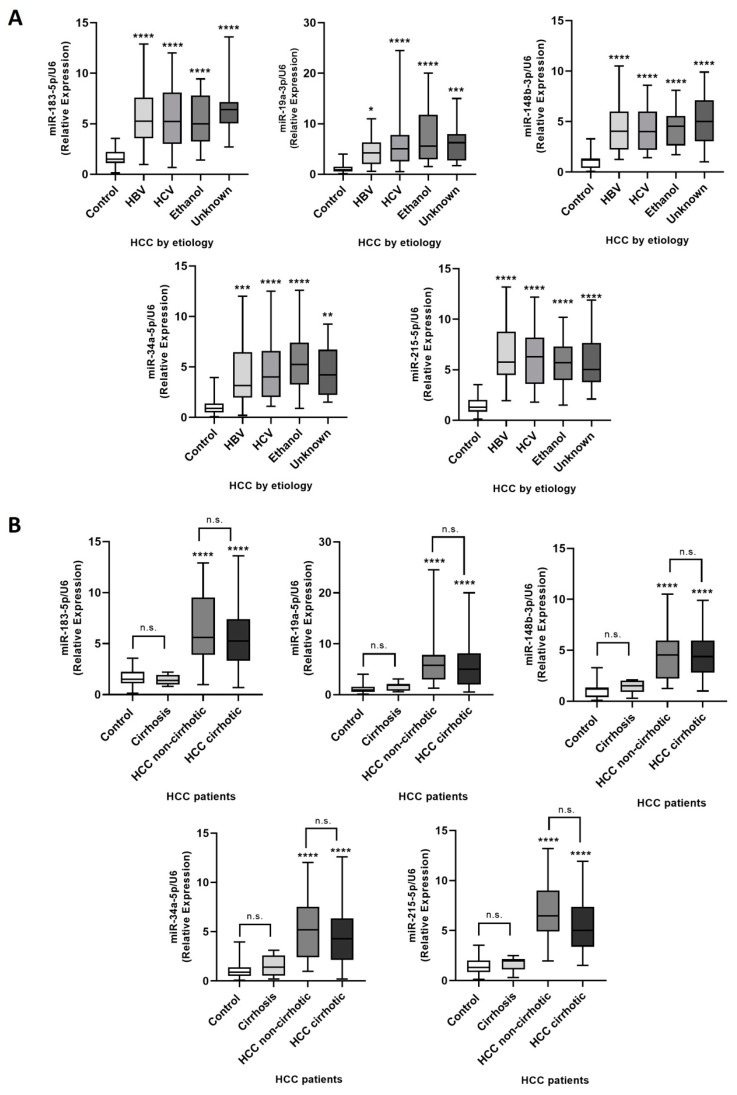
Analysis of miRNA expression in EVs circulating in the plasma of HCC patients with diverse etiologies. miRNA expression data by classifying patients according to HCC etiologies (**A**): HBV (*n* = 18), HCV (*n* = 35), ethanol (*n* = 25), and unknown (*n* = 12), as well as the presence (*n* = 60) or absence (*n* = 30) of cirrhosis (**B**) compared with healthy individuals (*n* = 41) or cirrhotic patients (*n* = 10). The miRNA expression levels were determined as relative values and normalized to U6 levels in both patient groups. The expression data are presented using box plots, showing the median and interquartile ranges. n.s. not significant **** *p* < 0.0001 vs. healthy subjects or cirrhotic patients, * *p* < 0.05, ** *p* < 0.01, *** *p* < 0.001.

**Figure 6 ijms-26-02563-f006:**
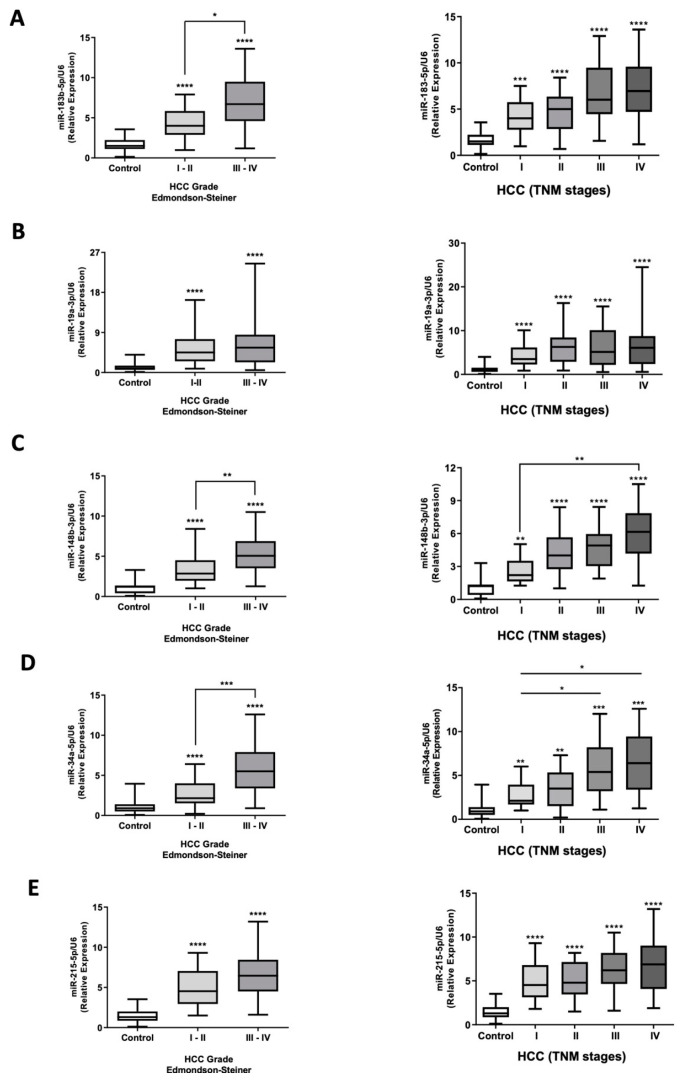
Expression assessments of miRNA contained in circulating EVs of HCC patients stratified by Edmondson–Steiner grading and TNM staging. EVs were purified by the plasma of HCC patients classified by Edmondson–Steiner grading I-II (*n* = 34) and III-IV (*n* = 56) and stratified by TNM staging I (*n* = 21), II (*n* = 21), III (*n* = 26), and IV(*n* = 22) and control subjects (*n* = 41). Total RNAs were obtained from the circulating EVs of plasma patients. (**A-E**) miRNA expression levels of plasmatic EVs of HCC patients stratified by tumoral grades (left) and stages (rigth) and healthy subjects were obtained by RT-qPCR. The miRNA expressions were calculated as relative expressions and normalized U6 levels in both patient groups. miRNA data expressions are represented in box plot graphs indicating the median and interquartile ranges. ** *p* < 0.01, *** *p* < 0.001, and **** *p* < 0.0001 vs. control subjects, * *p* < 0.05, ** *p* < 0.01, *** *p* < 0.001.

## Data Availability

All relevant data are included in the manuscript or Appendix A. Any additional request can be directed to the corresponding authors.

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
