# Peer review of "miRNAs-Set of Plasmatic Extracellular Vesicles as Novel Biomarkers for Hepatocellular Carcinoma Diagnosis Across Tumor Stage and Etiologies"

_ijms, 2025, doi:10.3390/ijms26062563_

Round 1
Reviewer 1 Report (New Reviewer)
Comments and Suggestions for Authors
The reviewed study by Dr. Molina-Pelayo and colleagues presents a multi-modal approach to microRNAs secreted via extracellular vesicles. By study design, 5 candidate miRNAs are discovered by a fluorometry based microarray with 84 cancer related miRNAs followed by validation in a DEN- treated wildtype rat model, hepatoma cell lines and a patients cohort.
My major concern is the one-dimensional, somehow biased conceptional presentation of primary liver cancer diagnostics and the role of liquid biopsy. In interdisciplinary teams comprising radiology, hepatology, surgery and pathology, diagnostic methods are not inadequate in my view (p.1, line 26 and p2 l.59-63). Especially contrast enhanced CT scans, as used in our center, have a high sensitivity and specificity (p2, l. 54) though uptake and washout dynamics. The limitation from our perspective is more, how patients can be motivated and enrolled in screening programs. I agree that liquid biopsies have a potential for screening and monitoring, but what is the explanation of circulating extracellular vesicles in non-metastazised HCC without angio- or lymphatic invasion (which can be TNM stage I and II)? Are these extracellular vesicles truely tumorous? Is known, if EVs derive from a paracrine, endocrine or exocrine secretion mode via bile acids? What does this mean for the systemic disease? And how does it affect longterm survivors after transplantation?
Minor concerns:
- the resolution of Figure 1 and 2 is too poor for a reasonable evaluation in my preliminary version
- p1 l. 25: primary liver cancer
- p1 l. 31 "human" patients? of course patients are human
- p4 l120 the microarray that was used to detect candidate miRNAs might be mentioned here in the legend, but it fabricate and platform are missing in material and methods
- p 5 l. 174: type of analyzed tissue is missing, I suppose plasma/serum?
- Figure 3: the term hepatocellular carcinoma HCC is excusively reserved for humans. Rodents terminology is equivocal: liver tumor or invasive liver tumor, if malignancy criteria are fullfilled
- Edmondson-Steiner grading was in many centers replaced by WHO grading
- p 12 l. 378 one word missing: hepatoma cell lines and DEN-Model?
- discussion: redundancy of results, too long
-
Author Response
Response to Reviewer #1
Dear Reviewer,
We would like to express our sincere gratitude for your constructive and positive comments. Your feedback has been very helpful for improvement of our work. We carefully reviewed and corrected the manuscript accordingly. The changes are indicated by line numbers and corresponding sections in the revised manuscript.
A detailed, item-by-item response to your comments is provided below this letter. In the revised manuscript, all changes are highlighted in yellow for clarity.
We look forward to hearing from you soon.
Kind regards,
Dr. Luis Castro-Sánchez
Dr. Enrique Chávez
EVALUATION
The reviewed study by Dr. Molina-Pelayo and colleagues presents a multi-modal approach to microRNAs secreted via extracellular vesicles. By study design, 5 candidate miRNAs are discovered by a fluorometry based microarray with 84 cancer related miRNAs followed by validation in a DEN- treated wildtype rat model, hepatoma cell lines and a patients cohort.
-My major concern is the one-dimensional, somehow biased conceptional presentation of primary liver cancer diagnostics and the role of liquid biopsy. In interdisciplinary teams comprising radiology, hepatology, surgery and pathology, diagnostic methods are not inadequate in my view (p.1, line 26 and p2 l.59-63). Especially contrast enhanced CT scans, as used in our center, have a high sensitivity and specificity (p2, l. 54) though uptake and washout dynamics. The limitation from our perspective is more, how patients can be motivated and enrolled in screening programs.
We greatly appreciate your valuable observations and suggestions. We fully agree that primary liver cancer diagnostics are conducted by interdisciplinary teams and that liquid biopsies hold potential for screening and monitoring. In this revised manuscript, we have removed statements suggesting that current diagnostic methods have low sensitivity and specificity for hepatocellular carcinoma (HCC) detection. Instead, we now refer to the lack of early screening and monitoring, as well as the limited availability of timely detection programs (p.1, line 26; p.2, lines 48-58), as major factors contributing to high mortality. Additionally, we have corrected the statement indicating that computed tomography and magnetic resonance imaging may be sufficient to confirm HCC. However, in many countries—particularly in developing regions—histopathological analysis of a liver biopsy, in addition to imaging methods, is still considered the definitive confirmatory test for HCC detection (p.2, lines 55-58). Finally, we clarify that there is a need for novel diagnostic methods for screening and monitoring that are non-invasive, sensitive, specific, and relatively easy to implement (p.2, lines 62-63).
-I agree that liquid biopsies have a potential for screening and monitoring, but what is the explanation of circulating extracellular vesicles in non-metastazised HCC without angio- or lymphatic invasion (which can be TNM stage I and II)?
We sincerely appreciate your insightful comments and questions. The tumor microenvironment (TME) plays a pivotal role in the onset and progression of HCC. The HCC progression is not solely determined by its intrinsic biological properties but is also significantly shaped by interactions with its surrounding TME. This microenvironment consists of a dynamic communication network of tumor cells and various adjacent cell types, including both innate and adaptive immune cells, endothelial cells, cancer-associated fibroblasts, vascular structures, and extracellular matrix components through cell-cell interactions, soluble factors and Extracellular Vesicles (EVs) [Liu YG, 2023].
EVs function as highly effective transporters of bioactive molecules such as proteins and nucleic acids (miRNA, long noncoding RNA, and DNA), lipids, and glycoproteins [Zeng Y, 2023]. These vesicles are crucial mediators of intercellular signaling, fostering communication between tumors cells and their surrounding TME. EVs derived from HCC cells establish a signaling network that drives tumor progression by facilitating key oncogenic mechanisms, including proliferation, apoptosis, angiogenesis, immune evasion, epithelial-mesenchymal transition, migration, invasion, the formation of pre-metastatic niches, and distant metastasis [Liu YG, 2024; Wang J, 2022; Wang C, 2023; Zhang Y, 2024; Lee, Y.T, 2021].
HCC-derived EVs frequently modulate tumor growth through both autocrine and paracrine signaling pathways. Since HCC is highly angiogenic, the extent of angiogenesis plays a critical role in disease prognosis. Additionally, EVs released by liver tumors can stimulate endothelial cells, promoting angiogenesis and thrombosis. Moreover, they can induce fibroblasts and bone marrow-derived mesenchymal stem cells to differentiate into myofibroblasts, further supporting the development of tumor-associated vasculature [Olejarz W, 2020]. It is well-documented that the molecular cargo within EVs enhances several tumorigenic mechanisms [Liu YG, 2024; Wang J, 2022; Wang C, 2023; Zhang Y, 2024; Lee, Y.T, 2021], including cell angiogenesis and proliferation. Consequently, the heightened angiogenic activity in HCC may contribute to its detection in systemic circulation.
EVs are present in circulation from the early stages of the disease and persist throughout its progression [Wang W, 2013; Van Niel, G, 2018]. Furthermore, their quantity is 3.0-fold higher in HCC cases compared to cirrhosis patients or 5.0-fold higher compared to healthy controls [Wang W, 2013; Julich-Haertel, H, 2017]. Finally, we propose that liver tumors begin secreting EVs in the early stages of the disease, actively driving HCC progression. Therefore, the detection of EVs in non-metastatic HCC represents a promising tool for screening and monitoring, potentially improving patient survival.
References
Liu YG, Jiang ST, Zhang L, Zheng H, Zhang T, Zhang JW, Zhao HT, et al. Worldwide productivity and research trend of publications concerning tumor immune microenvironment (TIME): a bibliometric study. Eur J Med Res. 2023;28:229. https://doi.org/10.1186/s40001-023-01195-3.
Zeng Y, Hu S, Luo Y, He K. Exosome cargos as biomarkers for diagnosis and prognosis of Hepatocellular Carcinoma. Pharmaceutics. 2023;15. https://doi.org/10.3390/pharmaceutics15092365.
Liu, YG., Jiang, ST., Zhang, JW. et al. Role of extracellular vesicle-associated proteins in the progression, diagnosis, and treatment of hepatocellular carcinoma. Cell Biosci 14, 113 (2024). https://doi.org/10.1186/s13578-024-01294-6.
Wang J, Wang X, Zhang X, Shao T, Luo Y, Wang W and Han Y (2022). Extracellular Vesicles and Hepatocellular Carcinoma: Opportunities and Challenges. Front. Oncol. 12:884369.doi: 10.3389/fonc.2022.884369.
Wang C, Zhang X, Yu J, Bu J, Gu X, Wang Y, Zhu X and Lin J (2023), Spotlights on extracellular vesicles in hepatocellular carcinoma diagnosis and treatment: an update review. Front. Bioeng. Biotechnol. 11:1215518.doi:10.3389/fbioe.2023.1215518.
Zhang Y, Zhang C, Wu N, Feng Y, Wang J, Ma L and Chen Y (2024) The role of exosomes in liver cancer: comprehensive insights from biological function to therapeutic applications. Front. Immunol. 15:1473030. doi: 10.3389/fimmu.2024.1473030.
Olejarz W, Kubiak-Tomaszewska G, Chrzanowska A, Lorenc T. Exosomes in angiogenesis and anti-angiogenic therapy in cancers. Int J Mol Sci. (2020) 21:1–14. doi: 10.3390/ijms21165840
Lee, Y.T.; Tran, B.V.; Wang, J.J.; Liang, I.Y.; You, S.; Zhu, Y.; Agopian, V.G.; Tseng, H.-R.; Yang, J.D. The Role of Extracellular Vesicles in Disease Progression and Detection of Hepatocellular Carcinoma. Cancers 2021, 13, 3076. https://doi.org/10.3390/cancers13123076.
Van Niel, G., D'Angelo, G. & Raposo, G. Shedding light on the cell biology of extracellular vesicles. Nat Rev Mol Cell Biol 19, 213–228 (2018). https://doi.org/10.1038/nrm.2017.125.
Wang W, Li H, Zhou Y, Jie S. Peripheral blood microvesicles are potential biomarkers for hepatocellular carcinoma. Cancer Biomarkers. 2013;13(5):351-357. doi:10.3233/CBM-130370.
Julich-Haertel, H.; Urban, S.K.; Krawczyk, M.; Willms, A.; Jankowski, K.; Patkowski, W.; Kruk, B.; Krasnodebski, M.; Ligocka, J.; Schwab, R.; et al. Cancer-associated circulating large extracellular vesicles in cholangiocarcinoma and hepatocellular carcinoma. J. Hepatol. 2017, 67, 282–292. https://doi.org/10.1016/j.jhep.2017.02.024.
-Are these extracellular vesicles truely tumorous?
EVs are secreted by both epithelial and mesenchymal cells during physiological and pathological processes. However, elevated levels of EVs have been reported in the blood of patients with several types of cancer, including HCC [Lee, Y.T, 2021]. Notably, EV levels have been shown to correlate with tumor size, pathological classification, and TNM stage [Wang W, 2013]. Following surgical tumor removal, the number of circulating EVs decreases seven days after curative R0 resection [Julich-Haertel, H, 2017]. Moreover, EV levels were significantly reduced in blood samples collected one month post-operatively compared to pre-operative samples [Wang W, 2013], suggesting that a significant portion of these vesicles originates from the tumor.
In our study, we identified differences in microRNA expression between hepatic tumor cell lines and non-tumor cell line. We further validated these findings in circulating EVs of DEN-induced rat liver cancer model and plasma samples of a cohort of patients. Our results indicate distinct expression patterns between HCC patients and healthy volunteers, suggesting that most of the microRNAs detected in our study originate from tumor-derived EVs. However, we do not exclude the possibility that some of these microRNAs may also originate from EVs secreted by cells surrounding the tumor or from other non-tumor-associated cell lineages.
Finally, although we did not directly confirm the tumoral origin of EVs in the patient cohort, our findings show that microRNAs associated with plasmatic EVs exhibit an increased expression levels in HCC patients compared to healthy or cirrhotic individuals. This suggests that these microRNAs may be associated with circulating EVs derived from the tumor.
References
Lee, Y.T.; Tran, B.V.; Wang, J.J.; Liang, I.Y.; You, S.; Zhu, Y.; Agopian, V.G.; Tseng, H.-R.; Yang, J.D. The Role of Extracellular Vesicles in Disease Progression and Detection of Hepatocellular Carcinoma. Cancers 2021, 13, 3076. https://doi.org/10.3390/cancers13123076.
Wang W, Li H, Zhou Y, Jie S. Peripheral blood microvesicles are potential biomarkers for hepatocellular carcinoma. Cancer Biomarkers. 2013;13(5):351-357. doi:10.3233/CBM-130370.
Julich-Haertel, H.; Urban, S.K.; Krawczyk, M.; Willms, A.; Jankowski, K.; Patkowski, W.; Kruk, B.; Krasnodebski, M.; Ligocka, J.; Schwab, R.; et al. Cancer-associated circulating large extracellular vesicles in cholangiocarcinoma and hepatocellular carcinoma. J. Hepatol. 2017, 67, 282–292. https://doi.org/10.1016/j.jhep.2017.02.024.
-Is known, if EVs derive from a paracrine, endocrine or exocrine secretion mode via bile acids? What does this mean for the systemic disease? And how does it affect longterm survivors after transplantation?
EVs have been detected in bile, where they originate from cholangiocytes and hepatocytes before being secreted into the bile canaliculi. Their influence on cholangiocyte and hepatocytes varies depending on their cellular source and whether they arise under normal or pathological conditions [Masyuk AI, 2010; Kostallari E, 2021]. Consequently, hepatocytes and cholangiocytes also sustain damage in cholestatic liver disease due to the progressive bile duct destruction, bile acid accumulation, and the continuous cycle of inflammation. [Hirschfield, G.M, 2010; Santiago, P, 2018].
Small EVs derived from bile interact with cholangiocyte primary cilia, leading to reduced phosphorylation of extracellular signal-regulated protein kinase 1/2 (ERK1/2), an upregulation of miR-15a, and the suppression of cholangiocyte proliferation [Masyuk AI, 2010; Kostallari E, 2021]. Additionally, EVs secreted by cholangiocytes play a role in cholestatic injury by transferring long-non-coding RNA (lncRNA) H19 to hepatocytes, which in turn downregulates small heterodimer partner (SHP) expression [Xiao Y, 2021]. EVs originating from cholangiocytes were swiftly internalized by Kupffer cells, where the transferred lncRNA H19 induced an upregulation of pro-inflammatory factors, such as IL-6 and chemokine (C-C motif) ligand 2 [Li, X, 2020]. A recent study demonstrated that EVs present in bile following partial hepatectomy (PH) enhance hepatocyte proliferation, accompanied by the upregulation of genes associated with cell cycle progression [Katsumi T, 2023]. All data suggests that there is a complex interrelationship mediated by EVs bile in the communication between liver cells.
In addition, the expression levels of miR-191, miR-486-3p, miR-1274b, miR-16, and miR-484 were significantly elevated in circulating EVs isolated from the bile of cholangiocarcinoma (CCA) patients compared to those obtained from individuals with bile leak syndrome, benign biliary obstruction, or primary sclerosing cholangitis [Li, L, 2014]. Although evidence suggests that EVs from bile can regulate various processes in both cholangiocytes and hepatocytes, their role in HCC development and their impact on long-term transplant survivors remain unclear. Additional research is required to better understand the specific roles bile EVs in the onset and progression of CCA and HCC.
References
Masyuk AI, Huang BQ, Ward CJ, Gradilone SA, Banales JM, Masyuk TV, Radtke B, Splinter PL, LaRusso NF. Biliary exosomes influence cholangiocyte regulatory mechanisms and proliferation through interaction with primary cilia. Am J Physiol Gastrointest Liver Physiol. 2010 Oct;299(4):G990-9. doi: 10.1152/ajpgi.00093.2010.
Enis Kostallari, Shantha Valainathan, Louise Biquard, Vijay H. Shah, Pierre-Emmanuel Rautou, Role of extracellular vesicles in liver diseases and their therapeutic potential, Advanced Drug Delivery Reviews, Volume 175, 2021, 113816, ISSN 0169-409X, https://doi.org/10.1016/j.addr.2021.05.026.
Xiao Y, Liu R, Li X, Gurley EC, Hylemon PB, Lu Y, Zhou H, Cai W, Long Noncoding RNA H19 Contributes to Cholangiocyte Proliferation and Cholestatic Liver Fibrosis in Biliary Atresia, Hepatology 70 (2019) 1658–1673. doi: 10.1002/hep.30698.
Katsumi T, Ishizawa T, Kobayashi T, Maki K, Suzuki F, Murakami R, Sato H, Ueno Y. Role of Bile-Derived Extracellular Vesicles in Hepatocellular Proliferation after Partial Hepatectomy in Rats. Int J Mol Sci. 2023 May 25;24(11):9230. doi: 10.3390/ijms24119230.
Hirschfield, G.M.; Heathcote, E.J.; Gershwin, M.E. Pathogenesis of cholestatic liver disease and therapeutic approaches. Gastroenterology 2010, 139, 1481–1496. doi: 10.1053/j.gastro.2010.09.004.
Santiago, P.; Scheinberg, A.R.; Levy, C. Cholestatic liver diseases: New targets, new therapies. Ther. Adv. Gastroenterol. 2018, 11, 1756284818787400. https://doi.org/10.1177/1756284818787400.
Li, X.; Liu, R.; Wang, Y.; Zhu, W.; Zhao, D.; Wang, X.; Yang, H.; Gurley, E.C.; Chen, W.; Hylemon, P.B.; et al. Cholangiocyte-derived exosomal lncRNA H19 promotes macrophage activation and hepatic inflammation under cholestatic conditions. Cells 2020, 9, 190. doi: 10.3390/cells9010190.
Li, L.; Masica, D.; Ishida, M.; Tomuleasa, C.; Umegaki, S.; Kalloo, A.N.; Georgiades, C.; Singh, V.K.; Khashab, M.; Amateau, S.; et al. Human bile contains microRNA-laden extracellular vesicles that can be used for cholangiocarcinoma diagnosis. Hepatology 2014, 60, 896–907. doi: 10.1002/hep.27050
Minor concerns:
-The resolution of Figure 1 and 2 is too poor for a reasonable evaluation in my preliminary version
Thank you for your valuable comments. In this version of the manuscript, we have adjusted the resolution of all figures to 300 dpi. Additionally, we have included a high-quality version of the figures in this response letter.
-p1 l. 25: primary liver cancer
Thank you very much for your suggestion. In this revised version, we have corrected the sentence (p.1, line 25).
-p1 l. 31 "human" patients? of course patients are human
Thank you very much for your comment. We have corrected these imprecisions throughout the manuscript.
-p4 l120 the microarray that was used to detect candidate miRNAs might be mentioned here in the legend, but it fabricate and platform are missing in material and methods
Thank you very much for your suggestion. In this revised version, we have added the name and manufacturer of the microarrays used for the expression analysis of candidate miRNAs (p.4, line 121). In addition, the technique for microarray-based miRNA expression analysis is described in the Materials and Methods section of the Supplementary Information (Page 5, data supplement), as including it in the main text would make the article excessively long due to the multiple techniques used in the experimental design.
-p 5 l. 174: type of analyzed tissue is missing, I suppose plasma/serum?
Thank you for your comment. Plasma was used as the sample for EV purification (p.5, line 175-176). Once purified, the EVs were used for Western blot and particle size analysis (p.6, line 176-177).
-Figure 3: the term hepatocellular carcinoma HCC is excusively reserved for humans. Rodents terminology is equivocal: liver tumor or invasive liver tumor, if malignancy criteria are fulfilled
We have replaced the term 'HCC' with 'liver tumor' when referring to the DEN-induced liver tumor rat model throughout the manuscript. Additionally, in Figure 3, the labels previously described as 'HCC' have been replaced with 'DEN'.
-Edmondson-Steiner grading was in many centers replaced by WHO grading
We greatly appreciate your valuable observations and suggestions. We fully agree that, in several centers, Edmondson-Steiner grading has been replaced by WHO grading. This system relies on H&E staining to assess tumor cell differentiation in comparison to normal hepatocytes. The WHO grading system shares many similarities with the Edmondson-Steiner system in terms of characterization; however, they do not completely overlap, particularly when morphologists adhere strictly to their respective criteria [Gisder DM, 2022].
Although histological grading of HCC plays a significant prognostic role, considerable heterogeneity persists in the microscopic assessment of this tumor. This inconsistency underscores the need for a standardized grading approach [Martins-Filho, 2017]. Consequently, a universally accepted grading system has yet to be established, with the Edmondson-Steiner system remaining one of the most widely used classification methods [Gisder DM, 2022; Martins-Filho, 2017; Pirisi M. 2010; Roberts DE, 2018]. In the hospital centers where we recruited patients for this study, the Edmondson-Steiner classification is still predominantly used, while WHO grading is employed to a lesser extent.
References
Gisder DM, Tannapfel A, Tischoff I. Histopathology of hepatocellular carcinoma - when and what. Hepatoma Res 2022;8:4. https://dx.doi.org/10.20517/2394-5079.2021.106.
Martins-Filho SN, Paiva C, Azevedo RS, Alves VAF. Histological Grading of Hepatocellular Carcinoma-A Systematic Review of Literature. Front Med (Lausanne). 2017 Nov 10;4:193. doi: 10.3389/fmed.2017.00193. PMID: 29209611; PMCID: PMC5701623.
Mario Pirisi, MD; Monica Leutner, MD; David J. Pinato, MD; Claudio Avellini, MD; Luca Carsana, MD; Pierluigi Toniutto, MD; Carlo Fabris, MD; Renzo Boldorini, MDArch Pathol Lab Med (2010) 134 (12): 1818–1822. https://doi.org/10.5858/2009-0551-OAR1.1
Roberts DE, Kakar S, Mehta N, Gill RM. A Point-based Histologic Scoring System for Hepatocellular Carcinoma Can Stratify Risk of Posttransplant Tumor Recurrence. Am J Surg Pathol. 2018 Jul;42(7):855-865. doi: 10.1097/PAS.0000000000001053.
-p 12 l. 378 one word missing: hepatoma cell lines and DEN-Model?
Thank you very much for your observations. We have changed the sentence to “we analyzed the expression of the five miRNAs commonly expressed in EVs derived HCC hepatoma cell lines in hepatic tissue and plasmatic EVs of a DEN-induced liver tumor model” (p.13, line 381-383).
-Discussion: redundancy of results, too long
We greatly appreciate your valuable scrutiny and recommendations. In this version of the manuscript, we have condensed the discussion and eliminated redundant results.

Reviewer 2 Report (Previous Reviewer 2)
Comments and Suggestions for Authors
The authors have answers may main concerns in a satifactory manner
Author Response
Response to Reviewer #2
Dear Reviewer,
We would like to express our sincere gratitude for your constructive and positive comments. Your feedback has been very helpful to our work. We carefully reviewed and corrected the manuscript accordingly. The changes are indicated by line numbers and corresponding sections in the revised manuscript.
A detailed, item-by-item response to your comments is provided below this letter. In the revised manuscript, all changes are highlighted in yellow for clarity.
We look forward to hearing from you soon.
Best regards,
Dr. Luis Castro-Sánchez
Dr. Enrique Chávez
EVALUATION
The authors have answers may main concerns in a satifactory manner
Thank you very much for considering our response satisfactory. Your suggestion has been extremely helpful for our work.

This manuscript is a resubmission of an earlier submission. The following is a list of the peer review reports and author responses from that submission.
Round 1
Reviewer 1 Report
Comments and Suggestions for Authors
Dear IJMS Editor and editorial team, I read the article by Molina-Pelayo FA. et al. (ijms-3307984) with attention, in particular, to the descriptive clinical part of the data analyzed in the 90 patients with HCC.
I would like to ask for the following clarifications in detail:
- Have all HCC patients received a diagnostic biopsy for HCC? Because only 32 had a diagnostic AFP test for HCC.
- Why do the Authors present a TNM classification, rather than a classification based on histotype, for example that of Edmondson-Steiner (ES): G1, well differentiate HCC from G4, indifferentiate HCC? Based on the ES degree of tumor differentiation, it was perhaps more interesting to capture tumor tissue-specific miRNA expressions, rather than the clinical stage of the tumor.
- Since most HCC cases were related to HCV infection, I ask:
Were these cases treated with antivirals?
Were they all viremic or had had a sustained virological response to therapy? I ask the same question for people with HBV infection. Since antiviral therapy, if done during the study presented, could influence a different profile of circulating miRNAs in EVs.
Can this statement be true?
The Figure 5, in the illustration of data reported in bottom letters F, G, H, is not well readable and needs to be optimized.
As regards the drafting of the paper in question, I found clarity and completeness in the methods described. The statistical analysis also seemed adequate to me.
Author Response
Response to Reviewer #1
Dear Reviewer,
We would like to express our sincere gratitude for your constructive and positive comments. Your feedback has been very helpful to our work. We have carefully reviewed and revised the manuscript accordingly. The changes are indicated by line numbers and corresponding sections in the revised manuscript, titled “New IJMS-3307984-manuscript.” and the supplementary information file named “New IJMS-3307984-supplementary”.
A detailed, item-by-item response to your comments is provided below this letter. In the revised manuscript and supplementary information, all changes are highlighted in yellow for clarity.
We look forward to hearing from you soon.
King regards,
Dr. Luis Castro-Sánchez
Dr. Enrique Chávez
EVALUATION
Dear IJMS Editor and editorial team, I read the article by Molina-Pelayo FA. et al. (ijms-3307984) with attention, in particular, to the descriptive clinical part of the data analyzed in the 90 patients with HCC.
I would like to ask for the following clarifications in detail:
- Have all HCC patients received a diagnostic biopsy for HCC? Because only 32 had a diagnostic AFP test for HCC.
Thanks for your valuable comments. We confirmed that all patients with HCC included in our study were accurately diagnosed using clinical parameters, with confirmation through liver biopsy and determination of tumor stage or grade (Lines 522 to 528). Additionally, we must clarify that, upon reviewing the data, we detected an error in the classification of patients with AFP levels lower or higher than 200 ng/mL in Supplementary Table 1. Specifically, 32 patients presented AFP levels ≤200 ng/mL, while 58 patients had levels >200 ng/mL.
It is important to emphasize that AFP levels ≤200 ng/mL do not exclude the presence of HCC. According to a meta-analysis of 51 studies, AFP levels in serum exhibit good accuracy for HCC diagnosis (Hang J, 2020); however, not all HCCs secrete AFP, and AFP may be elevated in cirrhosis or hepatitis cases. Therefore, relying solely on AFP is insufficient due to variability in sensitivity (41-65%) and specificity (80-94%) at different cohorts (Hang J, 2020 and Gupta S,2003). Furthermore, large multicentric survey showed that AFP-negative (<20 ng/mL) rates were found in 52% (261/502) patients with small HCCs (<3 cm), in 53.5% (51/95) patients at TNM stage I, in 48% (314/656) patients with Okuda stage 1, and in some advanced HCC patients [41.5% (24/58) at TNM stage IV and 28% (17/61) at Okuda stage 3] (Farinati F, 2006), indicating that nearly a half of HCC patients are AFP-negative, especially early and small HCC. Given the limitations of AFP as a standalone diagnostic marker, liver biopsy remains the gold standard for the definitive diagnosis of HCC.
References:
Hang J, Chen G, Zhang P, Zhang J, Li X, Gan D, Cao X, Han M, Du H, Ye Y. The threshold of alpha-fetoprotein (AFP) for the diagnosis of hepatocellular carcinoma: A systematic review and meta-analysis. PLoS One. (2020) 13;15(2):e0228857. doi: 10.1371/journal.pone.0228857.
Gupta S, Bent S, Kohlwes J. Test characteristics of alpha-fetoprotein for detecting hepatocellular carcinoma in patients with hepatitis C. A systematic review and critical analysis. Ann Intern Med. (2003) 1;139(1):46-50. doi: 10.7326/0003-4819-139-1-200307010-00012.
Farinati F, Marino D, de Giorgio M, Baldan A, Cantarini M, Cursaro C,et al. Diagnostic and prognostic role of alpha-fetoprotein in hepatocellular carcinoma: both or neither? Am J Gastroenterol. (2006) 101:524–32. doi: 10.1111/j.1572-0241.2006.00443.x
- Why do the Authors present a TNM classification, rather than a classification based on histotype, for example that of Edmondson-Steiner (ES): G1, well differentiate HCC from G4, indifferentiate HCC? Based on the ES degree of tumor differentiation, it was perhaps more interesting to capture tumor tissue-specific miRNA expressions, rather than the clinical stage of the tumor.
Your suggestion has been extremely helpful to our work. In response, we have incorporated a new figure and two tables into this revised version of the manuscript. These additions include the analysis of miRNA expression levels in EVs across different Edmondson-Steiner grades of HCC (grades I-II and III-IV) (Lines 222 to 234).
The five evaluated miRNAs demonstrated significant upregulation across all Edmondson-Steiner grades (Fig. 6) compared to healthy subjects (p < 0.0001). Notably, the expression levels of all analyzed miRNAs effectively discriminate HCC patients of any tumor grade from healthy individuals (Fig. 6A–E). Furthermore, the expression levels of miR-183-5p, miR-34a-5p, and miR-148b-3p showed a statistically significant difference across tumor grades (p < 0.0001) in HCC patients stratified by the Edmondson-Steiner grading system (Lines 222 to 234).
Additionally, we evaluated the diagnostic accuracy parameters in HCC patients stratified by Edmondson-Steiner grades; the results revealed sensitivity ranging from 79.41% to 92.86%, specificity 73.17% to 95.12%, accuracy of 80.00% to 93.26%, Youden’s Index of 0.6141 to 0.875, LR+ 3.29 to 87.50 while LR- was close to zero (Lines 280 to 284). Therefore, combined miRNAs contained in circulating EVs could increase their accuracy, and according to the American Joint Committee on Cancer (AJCC) and the International Union Against Cancer (UICC), the TNM staging system is widely recognized as an authoritative classification method and has been reported by some authors to outperform other staging systems (Edge S., 2010 and Wu J., 2005). Given the lack of significant differences in diagnostic accuracy parameters between TNM staging and Edmondson-Steiner grading, we conducted a combiROC analysis using the TNM stage to identify the optimal combination of EV-derived miRNAs for improving diagnostic accuracy (Lines 421 to 428).
References:
Edge, S.B.. AJCC Cancer Staging Manual; Springer, 2010; ISBN 9780387884400.
Wu, J.-C.; Huang, Y.; Chen, C.; Chang, T.; Chen, S.; Wang, S.; Lee, H.; Lin, P.; Huang, G.; Sheu, J.; et al. Evaluation of Predictive Value of CLIP, Okuda, TNM and JIS Staging Systems for Hepatocellular Carcinoma Patients Un-dergoing Surgery. J Gastroenterol Hepatol (2005), 20, 765–771, doi:10.1111/j.1400-1746.2005.03746.x.
- Since most HCC cases were related to HCV infection, I ask:
Were these cases treated with antivirals?
Thank you for your question. In our study, blood samples from HCC patients were collected at their initial presentation to the oncology departments of the participating medical centers. At the time of recruitment, none of the patients had received antiviral treatment for hepatitis B or C infection. To address this point more clearly, we have added this information to the manuscript (Lines 524 and 535).
Were they all viremic or had had a sustained virological response to therapy? I ask the same question for people with HBV infection. Since antiviral therapy, if done during the study presented, could influence a different profile of circulating miRNAs in EVs. Can this statement be true?
Thank you for your valuable question. As mentioned previously, blood samples were collected from patients upon their initial presentation to the oncology departments. At that time, none of the patients had received antiviral therapy for HBV or HCV infection. Consequently, all patients diagnosed with hepatitis virus infection were viremic and had not achieved a sustained virological response to therapy.
We agree that antiviral therapy could potentially influence the profile of circulating microRNAs (miRNAs) in extracellular vesicles (EVs), as it may impact viral replication and the immune response. Previous studies have reported that HCV/HIV co-infection does not alter the concentration or size of EVs but modulates the differential expression of miRNA cargo in plasma-derived EVs in patients undergoing antiviral treatment (Cairoli V., 2023). Additionally, HBV infection has been shown to modulate a distinct miRNA signature in HCC patients (Wang G., 2017). By including only patients who had not received antiviral therapy, we aimed to minimize this potential confounding factor in our study.
References:
Cairoli V, Valle-Millares D, Terrón-Orellano MC, Luque D, Ryan P, Dominguez L, Martín-Carbonero L, De Los Santos I, De Matteo E, Ameigeiras B, Briz V, Casciato P, Preciado MV, Valva P, Fernández-Rodríguez A. MicroRNA signature from extracellular vesicles of HCV/HIV co-infected individuals differs from HCV mono-infected. J Mol Med (Berl). (2023) 101(11):1409-1420. doi: 10.1007/s00109-023-02367-8.
Wang, G., Dong, F., Xu, Z. et al. MicroRNA profile in HBV-induced infection and hepatocellular carcinoma. BMC Cancer 17, 805 (2017). https://doi.org/10.1186/s12885-017-3816-1
The Figure 5, in the illustration of data reported in bottom letters F, G, H, is not well readable and needs to be optimized.
Thanks for your suggestions. To enhance the clarity and readability of the data, Figure 5 was divided into two separate figures. Considering that we included a new analysis based on the Edmondson-Steiner grading, as you previously suggested, we reorganized the figures as follows:
Figure 5: Panels A to E display the expression levels of the five miRNAs in HCC patients stratified by TNM staging.
Figure 6: Shows the expression levels of miRNAs in HCC patients classified according to Edmondson-Steiner grading.
Figure 7: Displays the combiROC analysis of the five miRNAs between HCC patients stratified by TNM staging and healthy subjects.
As regards the drafting of the paper in question, I found clarity and completeness in the methods described. The statistical analysis also seemed adequate to me.
Thank you for your valuable comments and contributions to this manuscript. We deeply appreciate your feedback, which has significantly improved our work.

Reviewer 2 Report
Comments and Suggestions for Authors
The manuscript by Molina-Pelayo describes their study on the potential use of plasma miRNA levels contained in EV in HCC diagnosis. The study examines miRNA in HCC cell lines, the DEN-induced HCC rat model and plasma samples of HCC patients. The study identified 5 miRNA as most promising noel biomarkers across the different models.
The aim the study is important and the design of the study is interesting, but is also a reason for concern. The main concern I have is that the majority of patients with HCC have cirrhotic livers, and also cirrhosis can undoubtedly have an effect on the release and cargo of EV. The controls used in the current study are selected as being healthy cells, healthy rat liver or plasma from healthy individuals. In my opinion this needs to be extended with additional controls that also reflect diseased livers and cirrhotic livers.
1. For the analysis of the cell lines a comparison of human hepatoma cells is made with the THLE cell line. All hepatoma cells are known to have completely different characteristics and the THLE cell line is of epithelial origin. Additional control cell lines should be included as comparison
2. The DEN-HCC rat model compares the HCC liver with healthy livers. Is it possible to include livers from DEN-induced rats prior to the development of HCC, so e.g. at week 12?
3. For plasma of HCC patients the comparison is made with healthy plasma. The comparison should be made with plasma from cirrhotic patients (since 67% of HCC are on a background of cirrhosis) and patients with other liver diseases resembling their etiology in a non-cirrhotic background. In this way the comparative parameters are better aligned.
4.
Author Response
Response to Reviewer #2
Dear Reviewer,
We would like to express our sincere gratitude for your constructive and positive comments. Your feedback has been very helpful to our work. We have carefully reviewed and revised the manuscript accordingly. The changes are indicated by line numbers and corresponding sections in the revised manuscript, titled “New IJMS-3307984-manuscript.” and the supplementary information file named “New IJMS-3307984-supplementary”.
A detailed, item-by-item response to your comments is provided below this letter. In the revised manuscript and supplementary information, all changes are highlighted in yellow for clarity.
We look forward to hearing from you soon.
King regards,
Dr. Luis Castro-Sánchez
Dr. Enrique Chávez
EVALUATION
The manuscript by Molina-Pelayo describes their study on the potential use of plasma miRNA levels contained in EV in HCC diagnosis. The study examines miRNA in HCC cell lines, the DEN-induced HCC rat model and plasma samples of HCC patients. The study identified 5 miRNA as most promising noel biomarkers across the different models.
The aim the study is important and the design of the study is interesting, but is also a reason for concern. The main concern I have is that the majority of patients with HCC have cirrhotic livers, and also cirrhosis can undoubtedly have an effect on the release and cargo of EV. The controls used in the current study are selected as being healthy cells, healthy rat liver or plasma from healthy individuals. In my opinion this needs to be extended with additional controls that also reflect diseased livers and cirrhotic livers.
We appreciated your valuable observations and suggestions. We understand your concern that most hepatocellular carcinoma (HCC) patients have underlying cirrhosis, which can influence the release and cargo of Extracellular Vesicles (EVs). Previous studies have demonstrated that the circulating miRNA signature has prognostic value for HCC risk in cirrhotic patients (Huang YH., 2017). Moreover, circulating miRNAs hold potential as valuable biomarkers for tracking the progression of liver diseases, spanning from cirrhosis to advanced HCC, and may also help in predicting patient responsiveness to first-line treatments such as sorafenib (D’Abundo L., 2024). The inclusion of patients with different liver pathologies, including cirrhosis, would provide new prognostic biomarkers for HCC risk. However, the aim of our study was to assess the diagnostic accuracy of miRNAs in discriminating between HCC patients regardless of etiology, and healthy individuals. Due to resource limitations and restricted access to samples from different liver pathologies, we did not include additional controls for comparison in our study. Recognizing this limitation, we have added the following statement to the discussion: “Finally, we understand as a limitation of our study that the miRNAs set only discriminates healthy individuals from HCC patients; therefore, future studies will consider evaluating the miRNA set in different liver pathologies” (Lines 466 to 468). Nevertheless, we are currently conducting new research aimed at assessing our panel of five miRNAs in cirrhotic patients without HCC to analyze the diagnostic accuracy for cirrhosis and the prognostic value for HCC risk.
References:
Huang, YH., Liang, KH., Chien, RN. et al. A Circulating MicroRNA Signature Capable of Assessing the Risk of Hepatocellular Carcinoma in Cirrhotic Patients. Sci Rep 7, 523 (2017). https://doi.org/10.1038/s41598-017-00631-9
D’Abundo, L., Bassi, C., Callegari, E. et al. Circulating microRNAs as biomarkers for stratifying different phases of liver cancer progression and response to therapy. Sci Rep 14, 18551 (2024). https://doi.org/10.1038/s41598-024-69548-4
For the analysis of the cell lines a comparison of human hepatoma cells is made with the THLE cell line. All hepatoma cells are known to have completely different characteristics and the THLE cell line is of epithelial origin. Additional control cell lines should be included as comparison
Thanks for your comments and suggestions. The THLE-2 cell line, derived from normal human liver epithelial cells preserves several hepatocyte-like characteristics, including the expression of cytokeratins, as well as albumin. Notably, THLE-2 cells are non-tumorigenic and possess functional cytochrome P450; epoxide hydrolase, NADPH cytochrome P450 reductase, superoxide dismutase, catalase, glutathione S-transferases, and glutathione peroxidase are also retained by THLE-2 (Pfeifer AM., 1993). In addition, THLE-2 has been widely used as non-tumoral human liver cell line (Giulitti F., 2021; Zhou, X., 2022; Mao Y., 2021; Koch DT., 2023; Huge N., 2022; Zhang B., 2022). Regarding to specify their origin (liver epithelial cells with hepatocyte-like characteristics), these were compared with hepatocellular carcinoma cell lines, providing a suitable baseline to assess malignant transformation and the progression of HCC.
Pfeifer AM, Cole KE, Smoot DT, Weston A, Groopman JD, Shields PG, Vignaud JM, Juillerat M, Lipsky MM, Trump BF, et al. Simian virus 40 large tumor antigen-immortalized normal human liver epithelial cells express hepatocyte characteristics and metabolize chemical carcinogens. Proc Natl Acad Sci U S A. 1993; 90(11):5123-7. doi: 10.1073/pnas.90.11.5123.
Giulitti F, Petrungaro S, Mandatori S, Tomaipitinca L, de Franchis V, D'Amore A, Filippini A, Gaudio E, Ziparo E and Giampietri C (2021) Anti-tumor Effect of Oleic Acid in Hepatocellular Carcinoma Cell Lines via Autophagy Reduction. Front. Cell Dev. Biol. 9:629182. doi: 10.3389/fcell.2021.629182
Zhou, X., Luo, J., Xie, H. et al. MCM2 promotes the stemness and sorafenib resistance of hepatocellular carcinoma cells via hippo signaling. Cell Death Discov. 8, 418 (2022). https://doi.org/10.1038/s41420-022-01201-3
Mao Y, Ding Z, Jiang M, Yuan B, Zhang Y, Zhang X. Circ_0091579 exerts an oncogenic role in hepatocellular carcinoma via mediating miR-136-5p/TRIM27. Biomed J. 2022 Dec;45(6):883-895. doi: 10.1016/j.bj.2021.12.009. (2021)
Koch DT, Yu H, Beirith I, Schirren M, Drefs M, Liu Y, Knoblauch M, Koliogiannis D, Sheng W, De Toni EN, Bazhin AV, Renz BW, Guba MO, Werner J, Ilmer M. Tigecycline causes loss of cell viability mediated by mitochondrial OXPHOS and RAC1 in hepatocellular carcinoma cells. J Transl Med. (2023); 21(1):876. doi: 10.1186/s12967-023-04615-4.
Huge N, Reinkens T, Buurman R, Sandbothe M, Bergmann A, Wallaschek H, Vajen B, Stalke A, Decker M, Eilers M, Schäffer V, Dittrich-Breiholz O, Gürlevik E, Kühnel F, Schlegelberger B, Illig T, Skawran B. MiR-129-5p exerts Wnt signaling-dependent tumor-suppressive functions in hepatocellular carcinoma by directly targeting hepatoma-derived growth factor HDGF. Cancer Cell Int. (2022); 22(1):192. doi: 10.1186/s12935-022-02582-2.
Zhang B, Zhou J. CircSEC24A (hsa_circ_0003528) interference suppresses epithelial-mesenchymal transition of hepatocellular carcinoma cells via miR-421/MMP3 axis. Bioengineered. (2022); 13(4):9049-9062. doi: 10.1080/21655979.2022.2057761.
The DEN-HCC rat model compares the HCC liver with healthy livers. Is it possible to include livers from DEN-induced rats prior to the development of HCC, so e.g. at week 12?
Thank you for your valuable comments. We agree that analyzing the expression of the five miRNAs prior to the development of HCC (such as in cirrhotic liver) in the rat model would provide valuable insights into the role of these miRNAs in the progression of HCC. However, since the aim of this study was to determine the miRNA expression levels across HCC cell lines, an HCC DEN-induced rat model, and plasma samples from patients, including additional time points such as the cirrhotic stage, was beyond the scope of our current experimental design. Due to limitations in generating and obtaining samples at this moment from a DEN model at the 12-week time point, we believe this would be an interesting prospect to explore in a future study.
For plasma of HCC patients the comparison is made with healthy plasma. The comparison should be made with plasma from cirrhotic patients (since 67% of HCC are on a background of cirrhosis) and patients with other liver diseases resembling their etiology in a non-cirrhotic background. In this way the comparative parameters are better aligned.
Thank you for your valuable comments. We agree that a high percentage of patients with cirrhotic livers develop HCC, and our study is no exception, as 66.6% of the patients in our study have cirrhosis. As we commented previously, including patients with different liver pathologies, such as cirrhosis, could provide additional prognostic biomarkers for HCC risk. However, the focus of our study was to assess miRNA diagnostic accuracy in distinguishing HCC patients from healthy individuals, regardless of the underlying liver pathology. Due to resource limitations and restricted access to diverse liver pathology samples, we did not include additional controls. We have recognized this limitation in the discussion section of our manuscript (Lines 466–468), suggesting that future studies will aim to include cirrhotic patients and patients with other liver diseases to further refine the diagnostic specificity of the miRNA set.

Round 2
Reviewer 2 Report
Comments and Suggestions for Authors
The authors did not address the issues raised. I am sure it is not too difficult to find collaborators who can provide matching samples from cirrhotic patients.
Author Response
Response to Reviewer #2
Dear Reviewer,
We would like to express our sincere gratitude for your constructive and valuable comments on our manuscript. A detailed response to your comments is provided below this letter.
King regards
Dr. Luis Castro-Sánchez
Dr. Enrique Chávez
EVALUATION ROUND 2
The authors did not address the issues raised. I am sure it is not too difficult to find collaborators who can provide matching samples from cirrhotic patients
We greatly appreciate your valuable observations and suggestions regarding our manuscript. We absolutely agree that including patients with cirrhosis could enhance our research and potentially identify novel diagnostic biomarkers for cirrhosis or prognostic biomarkers for HCC risk. However, incorporating cirrhotic patients into our study requires careful consideration of the following factors:
- First, while identifying collaborators to provide samples for such a study is not an issue, we face challenges related to existing ethical and procedural constraints. Our current protocol, approved by the ethics committees of the participating medical centers, specifies that plasma from HCC patients and healthy individuals are the only samples allowed to be analyzed for miRNA expression to identify potential diagnostic biomarkers for HCC. Unfortunately, patients with cirrhosis or other liver pathologies were not included in the original protocol approved by the Juarez Hospital of Mexico (approval no. CEI-HJM-544 P/014/2015) and the Ixtapaluca High Specialty Regional Hospital (approval no. NR-014-2022). To modify the protocol and include cirrhotic patients, we would need to submit an addendum for approval. This request would first need to be reviewed by the hospital committees, with the earliest consideration date being February 2025. Furthermore, the ethics committee must also approve the addendum, and the timeline for such approval is uncertain. If the addendum is approved by the ethics committee, the hospital committees would then require an estimated 4 to 6 months to formally authorize the modification.
- Second, including cirrhotic patients would require either approval of the addendum or the creation of a completely new protocol. If the addendum is rejected, a new protocol would need to be submitted for approval. It is critical to emphasize that informed consent from patients is a fundamental right, and it would be unethical to collect or use samples from cirrhotic patients without prior authorization. Therefore, we cannot collect or use such samples with no reviewed and approved protocol.
It is important to note that our study was designed to focus on miRNA expression levels in HCC patients compared to healthy individuals, as reported in previous studies (REFERENCES). The primary aim of our research is to evaluate the diagnostic accuracy of miRNAs in distinguishing HCC patients from healthy individuals, irrespective of etiology. In the first round of responses to your comments and suggestions, we recognized this limitation in our study and added the following statement to the discussion:
“Finally, we understand as a limitation of our study that the miRNA set only discriminates healthy individuals from HCC patients; therefore, future studies will consider evaluating the miRNA set in different liver pathologies” (Lines 466–468).
Finally, it is important to highlight that submitting a protocol for ethical review and collecting samples from cirrhotic patients would require a significant amount of time, as well as substantial human and economic resources. This process would take several months, or potentially up to a year. Despite the lack of these samples, we believe that the conclusions of this manuscript remain valid and are consistent with the results obtained. Nevertheless, we greatly appreciate your suggestions, which will guide our future research. We plan to conduct new studies aimed at evaluating our panel of five miRNAs in cirrhotic patients without HCC to analyze their diagnostic accuracy for cirrhosis and their prognostic value for HCC risk.
